

# Interpreting eddy covariance data from heterogeneous Siberian tundra: land cover-specific methane fluxes and spatial representativeness

Juha-Pekka Tuovinen[1], Mika Aurela[1], Juha Hatakka[1], Aleksi Räsänen[2,3], Tarmo Virtanen[2], Juha
Mikola[4], Viktor Ivakhov[5], Vladimir Kondratyev[6], Tuomas Laurila[1]

[1]Finnish Meteorological Institute, Climate System Research, P.O. Box 503, FI-00101 Helsinki, Finland
[2]Ecosystems and Environment Research Programme, Faculty of Biological and Environmental Sciences and Helsinki
Institute of Sustainability Science (HELSUS), P.O. Box 65, FI-00014 University of Helsinki, Finland
[3]Department of Geography, Norwegian University of Science and Technology, NO-7491 Trondheim, Norway
[4]Ecosystems and Environment Research Programme, Faculty of Biological and Environmental Sciences, University of
Helsinki, Niemenkatu 73, FI-15140 Lahti, Finland
[5]Voeikov Main Geophysical Observatory, St Petersburg, Russia
[6]Yakutian Service for Hydrometeorology and Environmental Monitoring, Tiksi, Russia

*Correspondence to*: Juha-Pekka Tuovinen (juha-pekka.tuovinen@fmi.fi)

**Abstract.** The non-uniform spatial integration inherent in the eddy covariance (EC) method provides an additional challenge for data interpretation when fluxes are measured in a heterogeneous environment, as the contribution of different surface types varies with flow conditions, potentially resulting in a bias as compared to the true areally averaged fluxes and surface attributes. We modelled flux footprints and characterized the spatial scale of our EC measurements at Tiksi, a tundra site in northern Siberia, including a comparison of different source area definitions. We used leaf area index (LAI) and land cover class (LCC) data, derived from very high spatial resolution satellite imagery and field surveys, and quantified the sensor location bias. We found that methane ($CH_4$) fluxes varied strongly with wind direction (from –0.09 to 0.59 $\mu$g m$^{-2}$ s$^{-1}$ on average), reflecting the distribution of different LCCs. Using footprint weights of grouped LCCs as explanatory variables for the measured $CH_4$ flux, we then developed a multiple regression model to estimate LCC-specific fluxes. This model showed that wet fen and graminoid tundra patches in locations with a high topography-based wetness index acted as strong $CH_4$ sources (0.95 $\mu$g m$^{-2}$ s$^{-1}$), while mineral soils were significant sinks (–0.13 $\mu$g m$^{-2}$ s$^{-1}$). Finally, to assess the representativeness of $CH_4$ flux measurements, we upscaled the LCC-specific fluxes to different spatial scales. This assessment showed that, despite the surface heterogeneity and rather poor representativeness of EC data with respect to the areally averaged LAI and coverage of some LCCs, the mean $CH_4$ flux measured during summer 2014 was close to the corresponding balance upscaled to an area of 6.3 km$^2$, with a location bias of 14 %. We recommend that EC site descriptions in a heterogeneous environment should be complemented with footprint-weighted high-resolution data on vegetation and other relevant site characteristics.



## 1 Introduction

Biosphere–atmosphere exchange of greenhouse gases (GHGs) is commonly measured using the micrometeorological eddy covariance (EC) method (Aubinet et al., 2012). This tower-based, non-intrusive technique provides spatially integrated flux data at the ecosystem scale with a typical integration domain of a few hectares. This is in stark contrast to flux chamber

measurements that can be focused on homogeneous small-scale ($< 1$ m$^2$) patches of an ecosystem or on individual plant communities (Livingston and Hutchinson, 1995; Virkkala et al., 2017). The spatial aggregation inherent in the EC data is a strong asset if one's objective is to study functioning or GHG exchange of an extensive, relatively homogeneous ecosystem. Heterogeneous landscapes consisting of a mosaic of differing vegetation and land cover patches, however, may entail issues on the interpretation of the spatial representativeness of measurements. This stems from the fact that the EC integration

process equals to non-uniform weighting of the upwind surface elements that influence the measured flux, thus potentially resulting in an unequal and temporally varying contribution from different surface cover types (Schmid, 2002). Especially isolated zones of high source/sink density may bias the estimated average flux of the area surrounding the EC tower. The spatial distribution of relative weights, a function that Leclerc and Thurtell (1990) coined "footprint", depends on the measurement height and strongly on wind direction. As the flux footprint is also affected by other properties of the

atmospheric flow, e.g., hydrostatic stability, directional averaging does not guarantee an unbiased flux estimate either.

Arctic tundra serves as a prime example of a surface that is heterogeneous with respect to biogeochemical processes. The vegetation, soil and land cover structure of tundra areas are fragmented, the landscape typically comprising patches of different plant communities, water bodies and other land cover types (Stow et al., 2004; Virtanen and Ek, 2014; Mikola et

al., 2018). Such heterogeneity concerns both the composition and configuration of surface properties. This is clearly manifested by the leaf area index (LAI), which shows a higher relative variation among sites in tundra than in any other biome (Asner et al., 2003), and there are also pronounced spatial and temporal LAI patterns at the landscape scale (Marushchak et al., 2013; Juutinen et al., 2017).

Surface heterogeneity can generate high variability in ecosystem–atmosphere methane (CH$_4$) fluxes (Olefeldt et al., 2013). Methane is a potent GHG that is accumulating in the atmosphere, and tundra biomes are responsible for about 3 % of the total CH$_4$ emissions estimated at 560 Tg yr$^{-1}$ (McGuire et al., 2012; Saunois et al., 2016). The emissions from tundra are predicted to increase substantially, as the permafrost soils contain vast amounts of organic carbon (approximately 1000 Pg in 0–3 m depth; Hugelius et al., 2014) and a fraction of this may be released into the atmosphere as a result of warming-induced

thawing of these soils (Schuur et al., 2015). While the processes involved are incompletely understood, with large uncertainties in the estimated increase in CH$_4$ production (Cooper et al., 2017; Mård et al., 2017), these emissions potentially enhance surface warming, thus creating a positive feedback to climate change (Schuur et al., 2015; Nzotungicimpaye and Zickfeld, 2017).




The heterogeneity in the land surface–atmosphere $CH_4$ flux originates from the multitude of biochemical and physical controls of the anaerobic $CH_4$ production and bacterial $CH_4$ oxidation (Whalen and Reeburgh, 1990; Lai, 2009; Bridgham et al., 2013). Methane can be released into the atmosphere through gradual diffusion in soil and water, in ebullition (bubbling) events and via plant-mediated advective transport. These processes involve different residence times and thus expose the produced $CH_4$ to different degree of oxidation. As a result of this complexity, field studies have identified a wide range of factors that are associated with the levels and variations of observed fluxes. Of these, soil temperature and moisture (or water table level) typically constitute the key environmental controls (Olefeldt et al., 2013). As a general rule, wet carbon-rich soils emit substantial amounts of $CH_4$, while dry tundra soils act as small net sinks (Lau et al., 2015); even if lesser in magnitude, the uptake flux may dominate the regional $CH_4$ balance (Jørgensen et al., 2015; D'Imperio et al., 2017). Methane flux strongly depends on vegetation and soil characteristics, such as the abundance of vascular plants with aerenchyma tissue facilitating gas transport; other important variables include substrate availability, and soil acidity and redox potential (Lai, 2009; Bridgham et al., 2013; Olefeldt et al., 2013). The key role of vegetation type was convincingly demonstrated by Davidson et al. (2016), whose simple vegetation classification could explain $CH_4$ emissions from Arctic tundra as accurately as the key environmental drivers.

Landscape heterogeneity not only calls for further long-term measurements of fluxes and their controls on multiple spatial scales but also necessitates development of techniques for data interpretation, including down- and upscaling methods required for generalization of observations. Micrometeorological models are available for estimating the flux footprint (Leclerc and Foken, 2014), and these models have been utilized in various ways when dealing with the representativeness of flux measurements, based on either chambers or EC, or a combination of the two. In its simplest form, such an analysis involves determination of footprint dimensions for typical flow conditions, to ensure that the expected "field of view" of EC measurements is sufficiently confined to the area of interest (e.g., Aurela et al., 2009), or to roughly evaluate the appropriate upscaling area of chamber measurements that are compared with concurrent EC data (e.g., Riutta et al., 2007).

Averaged footprints, or footprint "climatologies", can be calculated from time series of actual short-term (typically 30 min) meteorological data, providing a fuller view of the spatial extent of EC aggregation (e.g., Amiro, 1998). A flux-weighted footprint climatology has been suggested for further illustration, hinging on the assumption that the surface flux is spatially uniform within each short-term footprint (Chen et al., 2009). When combined with a land cover map, footprint time series can be used for data quality control by quantifying the contribution of different land cover types or, specifically, that of a certain ecosystem intended to be observed (Tuovinen et al., 1998; Rebmann et al., 2005; Göckede et al., 2008). The footprint function can also be used for a formal expression of the spatial, or more precisely the point-to-area (Nappo et al., 1982), representativeness of the EC measurements performed at a certain location. A suitable metric for this, termed the "sensor



location bias" by Schmid and Lloyd (1999), can be defined by comparing the footprint-weighted average of a surface-related quantity, mapped across the study area, to the corresponding arithmetic average.

While EC data from a heterogeneous environment are still commonly compared with plot-scale data without considering the differential weighting of the plots in the EC signal (e.g., Heikkinen et al., 2002; Sachs et al., 2010; Yu et al., 2013), footprint modelling has been combined with land cover information in various studies for a representative upscaling of chamber-based fluxes (e.g., Marushchak et al., 2016), plot-scale model results (e.g., Budishchev et al., 2014), remotely sensed fluxes (e.g., Chen et al., 2009) and vegetation data for model input (e.g., Stoy et al., 2013). The temporally varying footprint weights of different land cover types can also be taken as a basis for constructing statistical models to unravel land cover-specific flux estimates from the spatially aggregated EC data, but this depends on the quality of land cover data (Fan et al., 1992; Forbrich et al., 2011). The very high spatial resolution (VHSR) satellite imagery makes it possible to derive reliable land cover maps with as high as a ~1 m resolution. By utilizing such a detailed vegetation map, Budishchev et al. (2014) showed that footprint-weighting of modelled plot-scale $CH_4$ emissions from permafrost tundra resulted in a good agreement with EC measurements, while areally averaged fluxes failed to reproduce the heterogeneity-induced temporal variability. A similar conclusion was reached by Davidson et al. (2017), who upscaled chamber-based $CH_4$ fluxes for four sites in Alaska by means of VHSR vegetation maps.

The aims of the present study are threefold. First, we characterize the dimensions of the field of view and the point-to-area representativeness of the EC measurements carried out at a micrometeorological measurement station located on permafrost tundra at Tiksi in northern Siberia during summer 2014. We demonstrate and quantify the heterogeneity of this site, producing information that is essential for any further study exploiting these flux data. For this we employ a micrometeorological footprint model and detailed maps of surface characteristics, including land cover classes (LCCs) and LAI. These are based on VHSR satellite imagery and extensive field surveys, which still are scant for Siberian tundra; in addition, we utilize the topographic wetness index (TWI) derived from a VHSR digital elevation model. Second, we hypothesize that distinguishable mean fluxes can be determined for LCC groups that represent different $CH_4$ source/sink capacities; this can be accomplished by developing a multiple regression model that links these fluxes to the EC measurements via footprint weighting. This approach was motivated by the findings of Davidson et al. (2016) and the chamber measurements made at our site that showed that the effect of surface type was much larger than that of environmental controls (Vähä, 2016). Finally, the LCC-specific fluxes obtained in this way offer us an opportunity to upscale the $CH_4$ balance to the landscape scale and thus to evaluate the representativeness of EC measurements also with respect to $CH_4$ exchange.



## 2 Material and methods

### 2.1 Site and data

#### 2.1.1 Site description

The study area covers the surroundings of the micrometeorological GHG flux measurement station at Tiksi in northeastern
Russia. The EC tower of the station is located at 71.5943°N, 128.8878°E, 7 m above sea level, about 500 m from the
shoreline of the Laptev Sea, close to the Lena River delta. The site is run by the Finnish Meteorological Institute and is part
of the International Arctic Systems for Observing the Atmosphere (IASOA) activities (Uttal et al., 2016). The mean annual
temperature and precipitation at Tiksi in 1981–2010 were –12.7 °C and 323 mm, respectively (AARI, 2018). The study area
is within the continuous permafrost zone; the depth of the active soil layer typically varied within 0.2–0.4 m in early July –
mid-August 2014 (Mikola et al., 2018). The landscape around the EC tower represents the coastal tundra zone of eastern
Siberia with a high diversity of plant species and communities (Nyman, 2015; Juutinen et al., 2017; Mikola et al., 2018). The
terrain is relatively flat, sloping gently (2–3°) towards the south. This generates a hydrological gradient, and there is a small
brook running through the study area.

#### 2.1.2 Eddy covariance data

The $CH_4$ and energy fluxes used in the present study were measured continuously with the micrometeorological eddy
covariance method (Aubinet et al., 2012). The EC instrumentation consisted of a USA-1 (METEK GmbH, Elmshorn,
Germany) sonic anemometer/thermometer, an LI-7000 (LI-COR, Inc., Lincoln, NE, USA) $CO_2$/$H_2O$ analyser and an RMT-
200 (Los Gatos Research, Inc., San Jose, CA, USA) $CH_4$ analyser. The measurement height was 3 m. The sampling
frequency was 10 Hz, and the turbulent fluxes were calculated with the in-house PyBARFlucCalc program with 30 min
block averaging according to standard procedures, including double coordinate rotation, lag determination, wet-to-dry mole
fraction conversion and high-frequency flux loss correction where necessary (Aubinet et al., 2012). The $CH_4$ flux data were
screened for instationarity, weak turbulence (friction velocity < 0.12 m s$^{-1}$) and unphysical outliers. The $CH_4$ flux data
analysed in the present study cover the period of 5 July to 29 August 2014, which represents the thermal growing season of
that year, using the daily mean air temperature of 5 °C as the threshold.

#### 2.1.3 Surface characteristics mapping

The land cover classification was based on seven visually judged plant community types (PCTs) augmented by two non-
vegetated surfaces: (1) wet fen, (2) dry fen, (3) bog, (4) graminoid tundra, (5) flood meadow, (6) shrub tundra, (7) lichen
tundra, (8) bare ground and (9) water (Mikola et al., 2018). The PCTs were identified within an area of 1 km$^2$ around the EC
tower on the basis of an extensive vegetation and soil survey. They were verified using statistical ordination of the 92
established study plots according to plant species composition and functional plant and soil attributes. To extrapolate the



LCCs to the landscape scale (Fig. 1a, Fig. S1 in the Supplement), a supervised object-based random forest classification was carried out using two VHSR multispectral satellite images (12 August 2012 and 11 July 2015; WorldView-2, DigitalGlobe, Inc., Westminster, CO, USA) and a digital elevation model (DEM) constructed from the 2015 WorldView-2 stereo pair (Fig. 1b). The internal and external classification accuracy of the land cover classification were 80 and 49 %, respectively. For

details, see Mikola et al. (2018).

Using non-linear regression, the LAI of vascular plants was estimated from the normalized difference vegetation index (NDVI) calculated from the reflectance data of the 2012 WorldView-2 image, which represents the period of maximum LAI (Juutinen et al., 2017; Fig. 1c). The topographic wetness index (TWI) was calculated from the DEM using the method of

Böhner and Selige (2006) (Fig. 1d). TWI is defined as a function of the upslope contributing area and the local terrain slope, and thus it serves as a proxy for potential soil moisture. For details of the DEM and TWI data, see Mikola et al. (2018). All maps have a 2 m pixel size, and in this study they were limited to a circle with a radius of 1.4 km from the EC tower, which defines the domain of the present study. For upscaling to a regional scale, we also consider the LCCs determined within a larger area of 35.8 km$^2$ (Fig. S1 in the Supplement).

**2.1.4 Main features of the land cover classes**

The LCCs employed in the present study are described by Juutinen et al. (2017) and in greater detail by Mikola et al. (2018). Briefly, the dry fen, wet fen and bog classes represent peat-forming environments, while the other LCCs refer to surfaces that have no discernible peat. The vascular plant vegetation of fens is characterized by sedges (*Carex* spp.). *Sphagnum* and feather mosses are abundant in the dry fens, while the moss cover of wet fens is sparse. The bogs show microtopographic

variation, and their vegetation is dominated by dwarf shrubs, dwarf birch (*Betula nana*), and *Sphagnum* and feather mosses. The vegetation of flood meadows and graminoid tundra is dominated by graminoids (sedges and grasses) and willows (*Salix* spp.). The areas defined as shrub tundra have an abundant coverage of feather mosses and dwarf shrubs. In addition, lichen tundra patches with lesser biomass alternate with stony bare-ground surfaces.

In terms of soil properties of the vegetated surfaces, the dry fen, wet fen, bog and graminoid tundra LCCs stand out with their high organic matter (on average 38 % of soil dry mass) and water concentration (on average 73 % of fresh mass) in the top 10 cm soil layer, while the lowest concentrations (4 and 22 %, respectively) were found in the soils of the lichen tundra LCC (Mikola et al., 2018). The soil temperature at a depth of 15 cm was clearly highest in the lichen tundra sampling plots, and flood meadow and wet fen soils mostly had a higher temperature than those of the remaining community types. The

depth of the biologically active soil layer approximately doubled from early July to mid-August 2014 (the period when weekly measurements were taken). In mid-August, the active layer depth was highest, about 40 cm, at the wet fen and flood meadow plots and lowest, about 25 cm, at the shrub tundra and lichen tundra plots (Mikola et al., 2018).



## 2.2 Application of footprints

### 2.2.1 Footprint function

The footprint function, $\varphi$, defines the relationship between the source distribution of a quantity, $Q_\eta$, and the value of this quantity, $\eta$, at a certain location, $\boldsymbol{x}$. It is defined by the general integral equation (Pasquill and Smith, 1983; Schmid, 2002)

$$\eta(\boldsymbol{x}) = \int_\Omega \varphi(\boldsymbol{x}, \boldsymbol{x}')\, Q_\eta(\boldsymbol{x}')\, d\boldsymbol{x}', \tag{1}$$

where $\Omega$ denotes the integration domain and sinks are understood as negative sources. We assume that $\varphi$ only depends on the displacement of a source from the reference location, $\boldsymbol{x} - \boldsymbol{x}'$, and not on this location itself, implying that $\varphi$ is independent of $Q_\eta$; that is, the variable surface forcing is associated with inhomogeneities in a source strength of a passive scalar that do not affect the turbulence field. This is the so-called inverted plume assumption whose validity is a key prerequisite for the applicability of Eq. (1) (Schmid, 2002), which can be interpreted as a superposition of individual solutions for unit points

sources, if the equations governing $\eta$ are linear (Horst and Weil, 1994).

By specifying that the source term represents the horizontal distribution of a scalar flux density (hereafter referred to as "flux") at the ground surface and by defining the reference location $\boldsymbol{x}$ as the point at which the atmospheric vertical flux of this constituent is known, we can define the footprint function specific to this configuration, $f$, here expressed in polar

coordinates, with

$$\langle F \rangle = \int_0^\infty \int_0^{2\pi} f(\theta, r) F(\theta, r)\, d\theta dr, \tag{2}$$

where $F$ is the surface flux distribution, $\langle F \rangle$ is the vertical flux at a reference location above the surface, and $\theta$ and $r$ are the horizontal direction and distance with respect to this location. We assume that $f$ represents a normalized distribution; i.e., $f \in [0,1], \forall \theta, r$, and $\int_0^\infty \int_0^{2\pi} f(\theta, r)\, d\theta dr = 1$.

The flux footprint defined by Eq. (2) provides a means to identify and depict the influence of a two-dimensional surface on a point measurement. However, there are alternative ways of doing this, some of which are formalized here for clarity. Following Schmid (1994), we first define the cumulative footprint, or the source area, $\Pi$, in such a way that it corresponds to the smallest bounded region containing the surface elements that contribute to the measurement signal by a specified fraction $P \in (0,1)$. The source area defined in this way is bounded by a footprint isopleth, $f = f_P$, within which the integral of $f$

equals $P$,

$$\Pi(f_P) = \int_0^{2\pi} \left( \int_{r_{P,1}(\theta)}^{r_{P,2}(\theta)} f(\theta, r) dr \right) d\theta = P, \tag{3}$$



where the direction-dependent distances $r_{P,1}(\theta)$ and $r_{P,2}(\theta) > r_{P,1}(\theta)$ are defined by the condition $f > f_P$, if $r_{P,1} < r < r_{P,2}$. This formulation defines an arbitrary area, but it is also possible to derive averaged dimensions in terms of a direction-independent range $r_{P,1} - r_{P,2}$, i.e., an annulus, corresponding to a given $P$, if we first integrate along the angular coordinate,

$$\Pi^*(r_{P,1}, r_{P,2}) = \int_{r_{P,1}}^{r_{P,2}} \left( \int_{0}^{2\pi} f(\theta, r) d\theta \right) dr = P, \tag{4}$$

where the distances $r_{P,1}$ and $r_{P,2}$ minimize the annular area. (It is assumed here that the footprint distribution makes it
possible to define unique $r_{P,1}$ and $r_{P,2}$, which is not true for an arbitrary $f$). Both Eqs. (3) and (4) mean that also the closest edge of the source area, i.e., $r_{P,1}$, is located at a distance upwind from the measurement point. While Eq. (3) answers the question "*What is the area that contributes most to the measured flux?*", Eq. (4) corresponds to "*From which range does the flux originate, on average?*"

Alternatively, we can set $r_{P,1} = 0$ and define the cumulative footprint corresponding to Eqs. (3) and (4) as

$$\Pi_0(f_P) = \int_{0}^{2\pi} \left( \int_{0}^{r_P(\theta)} f(\theta, r) dr \right) d\theta = P \tag{5}$$

and

$$\Pi_0^*(r_P) = \int_{0}^{r_P} \left( \int_{0}^{2\pi} f(\theta, r) d\theta \right) dr = P, \tag{6}$$

respectively. In Eq. (5), the distance $r_P(\theta)$ coincides with a footprint isopleth. These definitions make it possible to characterize the measurement with a distance that is related to the traditional fetch concept (Dyer, 1963).

The area of influence is also commonly depicted on the basis of the cross-wind integrated footprint, where the integration is performed in Cartesian coordinates simply as

$$\Pi_C(f_P) = \int_{0}^{x_P} \left( \int_{-\infty}^{\infty} f(x, y) \, dy \right) dx = P, \tag{7}$$

where $x_P$ is the distance from the mast within which the proportion $P$ of the measured flux originates from, termed the "effective fetch" by Gash (1986). This is a useful definition if $f$ represents stationary conditions, i.e., a single short-term period, while $\Pi$, $\Pi^*$, $\Pi_0$ and $\Pi_0^*$ can be meaningfully determined also for the footprint averaged over multiple flow
conditions, i.e., for the time-averaged $f$ representing a footprint climatology.

### 2.2.2 Footprint-weighted averaging and sensor location bias

The footprint equation (Eq. 2) postulates that the flux at a certain location above the ground surface represents a spatial weighting of the surface flux distribution, where the weighting is defined by the footprint function $f$ that describes the



turbulent transport between each surface element and the reference point. In the context of EC measurements, the location of the EC tower defines the origin of the coordinate system $(\theta, r)$, and $f$ is specific to the sensor height and can be estimated by micrometeorological modelling (Leclerc and Foken, 2014); $\langle F \rangle$ denotes the measured flux, while typically $F(\theta, r)$ is unknown.

Based on $f(\theta, r)$, we can, analogously to Eq. (2), define footprint-weighted averages of other quantities. For a continuous variable $X$, such as LAI and terrain elevation, we write this average, or the "effective" value of $X$ related to a certain footprint $f$, as

$$\langle X \rangle = \int_0^\infty \int_0^{2\pi} f(\theta, r) X(\theta, r) \, d\theta dr. \tag{8}$$

We can also apply a similar averaging operation to an LCC map, in which each location (in practice, a pixel) is allocated to a single LCC. If we denote the LCC map by $\Lambda(\theta, r) = j$, where the integer $j = 1 \ldots N$ specifies the LCC at $(\theta, r)$, the weighted LCC corresponding to $f$ is defined as

$$\langle \Lambda \rangle_j = \int_0^\infty \int_0^{2\pi} f(\theta, r) \delta(\Lambda, j) \, d\theta dr, \tag{9a}$$

where

$$\delta(\Lambda, j) = \begin{cases} 0, & \Lambda(\theta, r) \neq j \\ 1, & \Lambda(\theta, r) = j \end{cases}. \tag{9b}$$

This provides the proportion of each LCC within the footprint, which can be calculated for a footprint climatology as well as a single footprint distribution. If the variable $X$ in Eq. (8) is LCC-specific but otherwise does not depend on location, i.e., we can specify constants $X_j$, $j = 1 \ldots N$, then we can combine Eqs. (8) and (9) to obtain the footprint-weighted $X$ as

$$\langle X \rangle = \sum_{j=1}^N \int_0^\infty \int_0^{2\pi} f(\theta, r) \delta(\Lambda, j) X_j \, d\theta dr. \tag{10}$$

To describe the point-to-area representativeness of the flux measurements with respect to a variable related to a surface property or exchange, we follow Schmid and Lloyd (1999) and define a metric that quantifies how well the measurement at a certain location reflects the actual conditions averaged over the area of interest. The sensor location bias for $X$ is calculated here as

$$\Delta_X = \frac{\langle X \rangle - \bar{X}}{\bar{X}}, \tag{11}$$

where $\bar{X}$ denotes the mean $X$ within the study area. This definition differs from the one introduced by Schmid and Lloyd (1999), who expressed the sensor location bias as $\Delta_X^2$. As $\langle X \rangle$ depends on the footprint and thus varies with time, $\Delta_X$ is not temporally invariant either.





We calculated the sensor location bias for terrain elevation, the maximum LAI and TWI that were mapped across the study area (Fig. 1). In addition, to investigate the effect of landscape heterogeneity, this bias was calculated for the mean $CH_4$ flux, for which the areal reference was obtained from the LCC-specific fluxes estimated with a multiple regression model (to be described in Sect. 2.3).

### 2.2.3 Footprint modelling

We calculated the flux footprints $f$ for each 30 min flux averaging period in a horizontal 2 m × 2 m grid by using the analytical footprint model developed by Kormann and Meixner (2001) (here "KM model"). The KM model is based on a stationary gradient diffusion formulation, building on the classical solution of the two-dimensional advection–diffusion equation with vertical power law profiles assumed for the mean wind speed and eddy diffusivity (Pasquill and Smith, 1983). As a novel feature, these profiles are related to the corresponding Monin–Obukhov similarity (MOS) profiles. The crosswind diffusion is assumed to be Gaussian and height-independent.

Our EC measurements provide the necessary input data for the KM model, including mean wind direction, $\theta$, mean horizontal wind speed (at anemometer height), $U$, friction velocity, $u_*$, hydrostatic stability, $L^{-1}$, and the standard deviation of lateral wind velocity, $\sigma_v$. When matching the wind and diffusivity power laws to the MOS profiles at the measurement height, the KM model does not require an explicit definition of roughness length, $z_0$, since the input data (i.e., $U$, $u_*$, $L^{-1}$) implicitly specify $z_0$ according to the MOS profile of the horizontal wind speed. In the case of a heterogeneous surface, this simplifies the computations significantly as compared to models that require additional flux aggregation procedures for estimating the effective $z_0$ (Göckede et al., 2006).

Independent of the flow conditions, a part of each footprint distribution formally extends beyond any finite target area. Therefore, in those footprint calculations that involve a surface property distribution such as the LCC map, we normalize the footprint integrated over the map area to 1, unless indicated otherwise. This means that the upper distance of radial integration in Eqs. (8)–(10) is set to a finite limit of $r_\mathrm{m}$ and the footprint-weighted averages are scaled by dividing by $\int_0^{r_\mathrm{m}} \int_0^{2\pi} f(\theta, r) \, d\theta dr$, where $r_\mathrm{m}$ ($\approx 1.4$ km) is the radius of the present surface maps.

### 2.2.4 Examples of flow conditions

To demonstrate how the EC flux measurement at Tiksi is affected by surface heterogeneity, we calculated with Eq. (8) the footprint-weighted averages of the surface attributes LAI, terrain elevation and TWI, and with Eq. (10) the footprint-weighted LCC areas of the nine classes shown in Fig. 1a. For this demonstration, we defined three flow situations in terms of the variables that affect the footprint in a given $\theta$, i.e., $U$, $u_*$, $L^{-1}$ and $\sigma_v$ (Table 1). These cases represent differing stability conditions, for which typical parameter combinations were derived from the measurement data employed in this study. The



$U - u_* - L^{-1}$ combination was constrained by $z_0 = 0.01$ m as calculated from the MOS profile of the horizontal wind speed (Pasquill and Smith, 1983). For lateral wind velocity fluctuations, we used the scaling $\sigma_v/u_* = 2.3$, which corresponds to the median of our data; for simplicity, this was adopted here for all stabilities.

## 2.3 Statistical model

### 2.3.1 Model formulation

Assuming that the $CH_4$ flux does not vary among the LCC-map pixels attributed to a certain LCC, we applied Eq. (10) and expressed each flux measurement as a weighted arithmetic mean of the LCC-specific fluxes $F_j$, $j = 1 \dots N$ (number of LCCs),

$$\langle F \rangle = \sum_{j=1}^{N} \langle \Lambda \rangle_j F_j, \tag{12}$$

where these fluxes are unknown, and the weights $\langle \Lambda \rangle_j$ (Eq. 9) are the fractional areas of the corresponding LCCs. Assuming further that the LCC-specific fluxes remain constant within a data set of $M$ (= 911) observations but the proportional LCC areas vary with the temporally changing footprint, we obtained a set of linear algebraic equations, from which a solution could be sought for $F_j$. We applied this idea by first defining aggregated LCC groups and formulated a linear regression problem as

$$\boldsymbol{Aq} = \boldsymbol{m} + \boldsymbol{e}, \tag{13}$$

where the matrix $\boldsymbol{A}$ $[M \times (N_A + 1)]$ consists of the proportional LCC areas of the aggregated LCCs for each observation, $\boldsymbol{q}$ $[(N_A + 1) \times 1]$ is a vector of the unknown parameters, $\boldsymbol{m}$ $[M \times 1]$ denotes the measurement vector, and $\boldsymbol{e}$ $[M \times 1]$ is the error term. $N_A$ denotes the number of those aggregated LCCs whose proportional area was included as an explanatory variable. This does not cover all the LCCs, and we included an intercept term in this regression equation so as to represent the remaining LCCs and the proportion of footprint extending beyond the study area; i.e., we did not scale the sum of $\langle \Lambda \rangle_j$ to 100 %.

We estimated $\boldsymbol{q}$ with the ordinary least square (OLS) estimator. Before calculating the standard errors of these estimates, we tested the model residuals for heteroskedasticity and serial correlation. Heteroskedasticity was tested with the White test that is based on an auxiliary regression, where squared residuals are regressed on original explanatory variables and their squares and cross products, and the inference is based on a Lagrange multiplier (LM) test statistic (Greene, 2012). Serial correlation was tested with the Breusch–Godfrey test, which is based on a similar LM principle where the OLS residuals are regressed on the original explanatory variables augmented by lagged residuals. If heteroskedasticity and serial correlation could not be ruled out, the standard errors for the model parameters were calculated with the Newey–West estimator, which is a robust estimator for the asymptotic covariance matrix of the OLS estimator (Greene, 2012). This would result in wider confidence



intervals than the traditional OLS-based standard errors. We assume that these estimates reflect the overall uncertainty emerging from measurement data, LCC classification and footprint modelling.

The agreement between the model and the observations was evaluated on the basis of the coefficient of determination, $R^2$,
root mean squared error, RMSE, and mean absolute error, MAE. The agreement was also examined as a function of wind direction, to verify that we can replicate the pronounced directional dependency of the observed fluxes (Aurela et al., 2015). The performance of the statistical model against independent data was assessed with 10-fold cross-validation (James et al., 2013).

### 2.3.2 Land cover class aggregation and upscaling of CH$_4$ fluxes

We hypothesize that mean CH$_4$ fluxes can be determined for LCC groups that are composed according to their expected source/sink capacity. This grouping was based on the documented vegetation and soil characteristics, reported in detail by Mikola et al. (2018) and Nyman (2015), and summarized here in Sect. 2.1.4. There are also data available from a limited set of flux measurements made with static chambers in summer 2014 that provide guidance for the LCC grouping (Vähä, 2016). In addition, we utilized the TWI map and defined areas of potentially wet soils as TWI > 4 (Fig. 1d). Using these data and
syntheses of CH$_4$ production and fluxes in similar ecosystems (Olefeldt et al., 2013; Nicolini et al., 2013; Turetsky et al., 2014; Lau et al., 2015; Petrescu et al., 2015; Treat et al., 2015) as background information, we defined four aggregated classes (Table 2, Fig. 2), for which the group-specific fluxes were determined with the statistical model described in Sect. 2.3.1.

The data sources listed above suggest that wet fens typically are strong CH$_4$ emitters, and thus the pixels of the wet fen LCC with TWI > 4 were selected for the first LCC group ("Strong source"). We also assumed that the pixels of the graminoid tundra LCC in the potentially wet locations should be included in this category as the graminoids at the site are dominated by aerenchymatous *Carex* spp. and *Eriophorum* spp. (Nyman, 2015), i.e., plants known to be associated with substantial CH$_4$ emissions. Other fens likely act as weaker emitters, so these are combined into another LCC group ("Moderate source"),
together with the bodies of freshwater (water LCC above the sea level). The previous data also justify an assumption that mineral soils, i.e., here the bare ground and lichen tundra LCCs, act as weak CH$_4$ sinks ("Sink"). The proportional areas of these three LCC groups were used as the explanatory variables in the regression model (Eq. 13), i.e., $N_A = 3$. The remaining pixels were allocated to the fourth group consisting of the LCCs that either are expected to have a very small CH$_4$ flux on average or cover only a limited area in flux footprints ("Neutral"). This group is included as an intercept in the regression
model, and we hypothesize that its estimated value is not statistically different from zero.

The CH$_4$ fluxes determined for the aggregated LCCs defined above were upscaled by a simple mosaic approach, i.e., by areal weighting of the group-specific fluxes. To illustrate how the upscaled flux depends on surface heterogeneity at different





spatial scales, the upscaling was performed for different sub-domains as a function of the distance from the EC tower, and also for a larger area of 35.8 km$^2$ (Fig. S1 in the Supplement).

## 3 Results and discussion

### 3.1 Demonstrating surface heterogeneity

To illustrate the expected range of flux footprint distributions of the EC measurements at Tiksi, Table 3 presents characteristic source area dimensions calculated with the meteorological data detailed in Table 1. Different source area definitions introduced in Sect. 2.2.1 are included: a full three-dimensional footprint, Π (Eq. 3), for a single, arbitrary wind direction; the corresponding cross-wind integrated footprint, $Π_C$ (Eq. 7); and an annular footprint climatology, $Π^*$ (Eq. 4), calculated assuming an equal frequency of wind directions (in which case $Π^*$ equals Π).

Table 3 demonstrates expected qualitative features of flux footprints:  the source areas are extensive over the aerodynamically smooth tundra terrain, and their dimensions increase with increasing atmospheric stability and with the proportional contribution to the measured flux, $P$ (Schmid, 1994). While the distance of maximal surface influence in a single footprint varies by a factor of 2 ($r_{max}$ = 18–35 m), depending on stability, for the estimated far end of the source area

contributing 90 % to the flux this factor is almost 20 ($r_{P,2}$ = 200–3500 m). For the annular climatology, the dimensions are somewhat smaller (Table 3). It is noteworthy that variation in the efficiency of horizontal diffusion also plays a marked role in the spatial weighting of different surface elements and that the results for the cross-wind integrated footprint are not identical to those for the three-dimensional function; however, they are similar to the dimensions of the annular climatology. If the integration for the climatology case originated from the EC tower ($Π_0$ and $Π_0^*$), the resulting distance would be very

close to $r_{P,2}$ (results not shown). Overall, these results indicate that, when reporting dimensions of the area of influence, it is important to state the definition adopted for this source area.

The 1.4 km radius of the circle centred at the EC tower, which defines our primary study area, was selected to result in a 95 % footprint coverage within this area in the neutral case. About 15 % of the footprint calculated for the stable flow example

extends beyond the limits of this area, while in the unstable case 99.7 % of the footprint is confined to the target circle (Fig. 3). The variation of the footprint-weighted LCC contributions, calculated with Eq. (10), as a function of wind direction demonstrates how the surface heterogeneity inherent in tundra landscape manifests itself in the EC measurement data (Fig. 3; see also Fig. S2 in the Supplement). As is obvious from the LCC map (Fig. 1a), there are large differences in the distribution of the contributing LCCs among different wind directions (Fig. 3). In the neutral case, for example, there are seven different

LCCs dominating at least in one sector. Turbulent mixing also plays a substantial role in the magnitude of relative LCC contributions, as the weighting of longer distances increases with increasing stability. In some directions, the contribution of



the most common LCCs is highly sensitive to atmospheric stability. In the northeast-to-east sector, for example, the rather limited dry fen patch located within a few tens of metres from the EC tower (Fig. 1a) contributes 45 % in the unstable case, while its contribution is only 13 % in the stable case (Fig. 3). Similarly, the relative importance of the extensive bare-ground area between the west and the north-east strongly depends on atmospheric stability.

The footprint-weighted surface characteristics, calculated with Eq. (8) for the cases detailed in Table 1, further demonstrate the heterogeneity-induced variations. The effective LAI originating from the footprint-weighting of the LAI map shows a strong dependency on wind direction: in the neutral case, for example, $\langle LAI \rangle$ ranges from 0.19 to 0.64 $m^2$ $m^{-2}$ (Fig. 4a). In the unstable case, the direction dependency is similar; however, the $\langle LAI \rangle$ values are up to 0.12 $m^2$ $m^{-2}$ lower than in neutral
conditions due to the dominance of bare ground in the vicinity of the EC tower in the north-western sector (Fig. 1a). As averaged over all directions, here assumed equally frequent, $\langle LAI \rangle$ is in all stability cases somewhat higher than the arithmetic areal average (Fig. 4a). Due to the directional variations in $\langle LAI \rangle$, the maximum sensor location bias ($\Delta_{LAI}$, Eq. 11) may exceed 90 % in the direction of the maximum $\langle LAI \rangle$ (Fig. 4d).

Based on a corresponding footprint-weighting, an effective mean value can also be determined for terrain elevation (Fig. 4b). This shows that, even though the topographic variability within the flux footprint is limited, slightly different terrain elevation patterns are associated with each flux measurement depending on both wind direction and stability. The sensor location bias for elevation is negative in almost all flow conditions, as the elevation is on average lower within the area that typically dominates the flux footprint (Figs. 1c and 4e). The area of predominantly bare ground is also apparent in the
effective TWI (Fig. 4c). In the east-to-south sector, the differences between the stability classes are due to the higher TWI values determined along the coast (Fig. 1d) that gain in importance in stable conditions. Between the south-west and the north-west, in contrast, $\langle TWI \rangle$ is higher in unstable conditions, which results from the more pronounced influence of the brook running nearby the EC tower. The footprint-weighted TWI averaged over all directions is, in all cases, close to the arithmetic area average (Fig. 4c), with the magnitude of the corresponding sensor location bias being lower than 30 % (Fig.
4f).

In addition to the examples presented above, we demonstrate the surface heterogeneity of the Tiksi landscape by calculating the mean LAI and LCC contributions from the time series of EC measurements adopted for the present analysis. Fig. 1a shows the footprint climatology for the growing season 2014, depicted as the cumulative flux according to the definition of
Eq. (3) (only the further distance visible). This source area is clearly asymmetric, and comparison with the data in Table 3 indicates that the source area for a certain $P$ is more limited than the corresponding area in typical neutral conditions; i.e., it effectively reflects slightly unstable conditions. Weighting the LAI distribution by the mean footprint results in a bias of $\Delta_{LAI} = 20.2$ %. This is much larger than the bias in the normalized vegetation difference index (NDVI) estimated for EC sites in northern China: at the 1 $km^2$ scale, this bias ranged from −6.9 to 4.2 % at eight sites with low vegetation, and even at a





land model scale of ~300 km$^2$ the mean absolute bias was not more than 6.5 % (Wang et al., 2016). Notwithstanding such a high degree of agreement, one of the sites was considered "disturbed". Kim et al. (2006) estimated an NDVI sensor location bias of less than 4 % for both an oak/grass and a slash pine site at the 1 km$^2$ scale and concluded that the related EC measurements were unbiased, while a bias of 28 % determined for a grassland site was considered problematic.

For most of the LCCs at Tiksi, the field of view of the EC sensors averaged over the growing season clearly differs from the areal coverage of the LCC within the study area (Table 4; see also Fig. S3 in the Supplement for the effect of wind direction and stability). The difference is largest for the water LCC, the pixels of which are concentrated on the fringes of the study area, and for the flood meadow category with a limited coverage, but there are also major differences among the dominating

terrestrial classes, such as shrub tundra and wet fen; the surface elements attributed to these LCCs contribute to the EC observations less (by 37 %) than their total areal coverage would suggest. Rescaling the proportions after removing the marine water pixels from the study area for their limited footprint weight further amplifies this difference.

If the areal LCC proportions are calculated within the non-circular area defined by the 90 % cumulative footprint (Eq. 3, Fig.

1a), some of these proportions change dramatically (Table 4). We also included in the comparison the LCC distributions for a circle with a radius of 1 km, which halves the study area, and for a 35.8 km$^2$ area (Fig. S1 in the Supplement). Compared to the latter, the study area has a similar coverage of fens, bare ground and lichen tundra; water and shrub tundra surfaces are under-represented, and bogs and graminoid tundra are over-represented. Overall, these results demonstrate the multiscale heterogeneity of the site and indicate that here the representativeness cannot be described as a proportional coverage of a

single target LCC in the footprint climatology, as is the case for most EC sites (Göckede et al., 2008).

**3.2 Land cover-specific fluxes**

The parameters of the regression model introduced in Sect. 2.3.1 were estimated with OLS for the LCC aggregation presented in Sect. 2.3.2. This produced model residuals that exhibited both heteroscedasticity and autocorrelation (White LM test statistic $MR^2 = 92 > \chi^2_{0.99(9)}$, Breusch–Godfrey LM test statistic $(M-1)R^2 = 117 > \chi^2_{0.99(1)}$). Thus the confidence

intervals were based on the heteroskedasticity and autocorrelation consistent Newey–West estimator. Even when these (larger) confidence intervals were introduced, all the estimated parameters except for the constant, i.e., those representing aggregated LCCs with expected CH$_4$ exchange, proved to be statistically different from zero ($p < 0.05$; Table 5). The results were also in perfect accord with our qualitative hypothesis on CH$_4$ flux variability among the LCCs: the model could differentiate between the high emitters, moderate emitters and sinks without any explicit prior information on this pattern.

The temporal variation of the estimated 30 min ecosystem-scale CH$_4$ fluxes is consistent with observations, even though it is obvious that the full range of variability, most notably the peak values, cannot be reproduced ($R^2 = 0.797$, RMSE = 0.0994




µg m$^{-2}$ s$^{-1}$, MAE = 0.0686 µg m$^{-2}$ s$^{-1}$). However, part of this variation arises from measurement noise, and in this context it is crucial that the mean fluxes are modelled accurately also when considering the strong wind direction dependence of observations (Fig. 5). This dependence, obviously generated by the systematic LCC variations within the flux footprint (Fig. 3), is a key pattern in this data set and must be taken into consideration when calculating representative CH$_4$ balances
(Aurela et al., 2015). The model residuals differ significantly ($p < 0.05$) from zero only in a narrow southeastern wind sector, where the model slightly overestimates. The 10-fold cross-validation statistics show that the model performs against independent data only marginally worse than the fit to the full data set ($R^2$ = 0.794, RMSE = 0.1000 µg m$^{-2}$ s$^{-1}$, MAE = 0.0691 µg m$^{-2}$ s$^{-1}$).

The LCC group-specific fluxes are in accordance with the extensive synthesis of chamber-based CH$_4$ flux measurements across permafrost zones conducted by Olefeldt et al. (2013). This database indicates that the mean flux at peatland sites during the growing season ranged from 0.03 µg m$^{-2}$ s$^{-1}$ on dry tundra to 0.75 µg m$^{-2}$ s$^{-1}$ on wet tundra (medians of site-specific fluxes). The mean flux at the sites classified as permafrost fen was within 0.48–1.70 µg m$^{-2}$ s$^{-1}$ at 50 % of the sites. The micrometeorological measurements that integrate over the surface heterogeneity of tundra landscape typically show
lower fluxes. For example, Sachs et al. (2008) measured a mean CH$_4$ emission of 0.22 µg m$^{-2}$ s$^{-1}$ from polygonal tundra in the nearby Lena River delta, which is close to our mean flux. Eight other Arctic tundra sites in Siberia, Alaska and Greenland had a mean summer flux within the range of 0.12–1.39 µg m$^{-2}$ s$^{-1}$ (Parmentier et al., 2011; Tagesson et al., 2012; Castro-Morales et al., 2017).

The sink efficiency estimated for the mineral soil LCCs  (–0.131 µg m$^{-2}$ s$^{-1}$) seems high when compared to previous data (Turetsky et al., 2014; Lau et al., 2015; Jørgensen et al., 2015; D'Imperio et al., 2017). However, this estimate is consistent with the measured EC fluxes and thus not an artefact of the modelling procedure. This can be observed by inspecting the cases in which the proportion of the assumed sink LCCs in the flux footprint exceeds 80 % (within the wind direction sector of 330–360°). By ignoring the other LCCs, we obtain an apparent mean CH$_4$ flux of –0.109 µg m$^{-2}$ s$^{-1}$ for these cases, while
the corresponding modelled (for all LCCs) and measured fluxes are –0.093 and –0.094 µg m$^{-2}$ s$^{-1}$, respectively. Furthermore, 32 chamber measurements were conducted on bare ground at the site in summer 2014, yielding a mean of –0.12 µg m$^{-2}$ s$^{-1}$ (Vähä, 2016).

    Our results are necessarily influenced by the quality of the land cover classification. The accuracy assessment indicates that
especially the flood meadow LCC is poorly classified (Mikola et al., 2018); however, this LCC only appears along the brook and has a very limited coverage. More importantly, the dry fen, wet fen and graminoid tundra pixels may be partly mixed up. The field data and multivariate data analysis of Mikola et al. (2018) indicate that the variation in plant functional type composition within these LCCs indeed overlap, which impairs the separation between the strong and moderate source LCC groups. On the other hand, the large areas of bare ground and lichen tundra with low organic soil content, i.e., the assumed



CH$_4$ sink areas, could be identified reliably (Mikola et al., 2018). Despite the uncertainties, the classification of vegetation types allows us to meaningfully group the surface elements also according to their CH$_4$ exchange potential. This relationship shows that vegetation type reflects the integrated effect of a range of processes that control net production and efflux of CH$_4$, such as the availability of substrates and gas transport routes (Davidson et al., 2016, 2017). Thus a vegetation classification

based on VHSR satellite imagery provides us with a straightforward means of upscaling the average LCC-specific fluxes without considering environmental controls.

**3.3 Upscaled CH$_4$ fluxes**

By upscaling the mean CH$_4$ fluxes estimated for the LCC groups, we estimated the effect of EC tower location on the spatial representativeness for the mean CH$_4$ flux observed during the growing season of 2014 (0.208 µg m$^{-2}$ s$^{-1}$). In other words,

adopting the data shown in Table 5 as a reference for the CH$_4$ flux averaged over the study area, we could calculate the sensor location bias for CH$_4$ flux (Fig. 5) similarly to the results shown in Sect. 3.1 for LAI, terrain elevation, TWI and LCC proportions. As the relative area of the coastal waters is significant within the study area but minor in the average flux footprint (Fig. 1, Table 4), these water areas were excluded from the upscaling domain.

Calculating the sensor location bias for CH$_4$ flux equals to a linear transformation of the observed fluxes. Thus the pronounced directional dependence of CH$_4$ fluxes translates into an equally pronounced variation in this bias estimate, which ranges approximately from –200 to 400 % (Fig. 5). The bias is smallest in eastern and western wind directions. However, the effective LCC composition is very different in these directions, with a much smaller coverage of fens in the west (Fig. 3).

The areally averaged CH$_4$ flux depends on the upscaling domain in a non-monotonous manner (Fig. 6). Defining the

reference area as a function of the radius of a circular area centred at the EC tower, the magnitude of sensor location bias is less than 30 % for the distances exceeding 80 m, and less than 10 % for the distances of ca. 640–1350 m. Acknowledging the statistical uncertainty in the upscaled fluxes, determined from the LCC-specific uncertainty estimates (Table 5), the measured mean flux is within the 95 % confidence interval for distances larger than 605 m. For the primary study area, the mean bias during the growing season is 13.9 % and the corresponding 95 % confidence interval is [–0.3 %, 32.9 %].

While formally the overestimation of EC measurements of the CH$_4$ flux averaged over the study area is not statistically significant ($p > 0.05$), the sensor location bias could be minimized by reducing the radius to 800–1000 m. Considering the CH$_4$ fluxes alone, the LCC distribution shown for the area extending to 1 km from the EC tower would be a suitable choice for a representative site description. Even if the coverage of the nine basic LCCs clearly differs from their footprint-weighted

contributions (Table 4), the four classes aggregated according to their assumed CH$_4$ emission potential cover areas rather similar to those within the original study domain (Table 6). Within the regional upscaling area of 35.8 km$^2$, the strong emitters are less common, but the total flux is only 13 % lower than within the original study area. On the other hand, freshwater bodies occupy a larger relative area (Table 4). These were included here in the moderate source LCC group, but



the actual emissions from these surfaces could not be estimated as their total area within the flux footprint is minute. Nevertheless, there is an increasing amount of evidence that Arctic lakes and ponds may emit significant amounts of $CH_4$ (Wik et al., 2016). At all scales, it is necessary to allow for the sink areas that play a significant role in the upscaled balance. However, the agreement of $CH_4$ fluxes between different scales may be considered somewhat fortuitous and implies little

about carbon dioxide and other scalar fluxes that have different spatial patterns. For example, Chen et al. (2009) estimated a sensor location bias of up to 55 % for the monthly gross primary production of a Douglas fir-dominated forest.

## 4 Conclusions

The eddy covariance flux measurement technique is commonly considered to have an advantageous spatial averaging property, sometimes to the extent that it is assumed to "provide an accurate integration of the overall flux from the

[heterogeneous] ecosystem" (Turner and Chapin III, 2006). However, this notion is limited and potentially misleading as a universal premise, since this integration process involves differential and temporally varying weighting, a well-known feature of EC measurements, which we in the present study demonstrated and quantified for a heterogeneous tundra site in northeastern Russia. The $CH_4$ fluxes measured at Tiksi were highly variable due to the variation in vegetation composition and soil wetness within the tundra landscape around the EC tower. During summer 2014, the mean bias of observations with

respect to the upscaled flux varied strongly with wind direction, ranging from –200 to 200 %,

By combining VHSR satellite imagery and footprint modelling, we could statistically estimate the contribution of the main land cover types to EC measurements. Methane emissions mainly originated from wet fen and graminoid tundra patches in locations with topography-enhanced soil wetness, where conditions are favourable for $CH_4$ production and efflux (mean flux

0.95 µg m$^{-2}$ s$^{-1}$ in summer 2014). Another noteworthy feature is that the areas of bare soil and lichen tundra acted as strong $CH_4$ sinks (–0.13 µg m$^{-2}$ s$^{-1}$). Despite the surface heterogeneity and directional variations in the point-to-area representativeness of EC measurements, the mean $CH_4$ flux measured during this season can be considered unbiased, and even more so if the present area of interest is halved, i.e., considered to extend up to 1 km from the EC tower. On the other hand, the measured fluxes overestimate the regional (35.8 km$^2$) balance by 30 %.

An important implication emerging from our results concerns the definition of the study area. It is customary to report a "site description" that documents the key ecological characteristics of the area of interest. Within a homogeneous environment, collating the necessary site data is straightforward in terms of statistical representativeness because the outcome is insensitive to the spatial sampling design. Furthermore, the representativeness of EC measurements can be simply assessed

by considering the coverage of a single target LCC within the flux footprints. In heterogeneous environments, however, there is a risk for a serious mismatch between the EC flux measurements and the site data, even in cases of an unbiased description of the study area. Our results show that the land cover type composition sampled by the EC measurement was significantly



different from the actual LCC coverage within our study area, which as such was chosen to be consistent with the dimensions of a typical flux footprint and considered characteristic of the landscape. There were large differences in the weighting of the LCCs also when averaged over a full growing season. Thus the small-scale heterogeneity was so high as to result in rather unfavourable representativeness metrics for key surface features such as LAI and LCC fractions, even though the sampled LCC distribution proved to be representative in terms of the mean $CH_4$ flux. These findings suggest that it is beneficial to present a more integrated site and flux data description than what has been considered standard, i.e., to also include data on footprint-weighted surface attributes data and point-to-area representativeness.

In a follow-up study, we will investigate annual $CH_4$ flux data from Tiksi to better understand environmental controls and to derive long-term $CH_4$ balances. We assume that estimation of LCC-specific fluxes, accomplished here with a regression model, provides a new avenue to filling the inevitable gaps in the measurement data time series. This proposition is supported by the good out-of-sample validation statistics presented, as holding the validation data out during parameter estimation is equivalent to generating missing data that need to be gap-filled. This kind of an approach is potentially applicable to those data gaps that are related to the gas concentration measurement, for example due to malfunctioning of the gas analyser, i.e., gaps that appear in the $CH_4$ flux data but not in the momentum and sensible heat fluxes. Furthermore, we anticipate that flux sites with more than one EC tower provide new opportunities for the estimation of LCC-specific fluxes; more advanced inverse modelling techniques should be explored for this.

**Acknowledgements**

This study was financially supported by the Academy of Finland, projects "Greenhouse gas, aerosol and albedo variations in the changing Arctic" (project no. 269095), "Carbon balance under changing processes of Arctic and subarctic cryosphere" (project no. 285630), "Constraining uncertainties in the permafrost-climate feedback" (project no. 291736) and "Carbon dynamics across Arctic landscape gradients: past, present and future" (project no. 296888); the European Commission, project "Changing permafrost in the Arctic and its global effects in the 21st century (PAGE21)"; and the Nordic Council of Ministers, DEFROST Nordic Centre of Excellence within NordForsk.




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





Table 1. Flow conditions assumed for the example calculations.

| Case | $L^{-1}$ (m$^{-1}$) | $u_*$ (m s$^{-1}$) | $\sigma_v$ (m s$^{-1}$) | $U$ (m s$^{-1}$) |
|---|---|---|---|---|
| Unstable | −0.2 | 0.15 | 0.35 | 1.8 |
| Neutral | 0 | 0.40 | 0.92 | 5.7 |
| Stable | 0.1 | 0.10 | 0.23 | 1.8 |

Table 2. Aggregated land cover classes for the regression model.

| LCC group description | LCCs included |
|---|---|
| Strong source | Wet fen, TWI > 4 |
| | Graminoid tundra, TWI > 4 |
| Moderate source | Wet fen, TWI ≤ 4 |
| | Dry fen |
| | Water, above sea level |
| Sink | Bare ground |
| | Lichen tundra |
| Neutral | Other |





Table 3. Source area dimensions for the example cases specified in Table 1, calculated according to different definitions: (1) Single three-dimensional footprint $\Pi$, Eq. (3); (2) Single cross-wind integrated footprint $\Pi_C$, Eq. (7); and (3) Annular footprint climatology $\Pi^*$, Eq. (4).

| Case[a] | | $P = 25\ \%$ | $P = 50\ \%$ | $P = 75\ \%$ | $P = 90\ \%$ |
|---|---|---|---|---|---|
| **(1) Single three-dimensional ($\Pi$)** | | | | | |
| | $r_{max}$ (m) | | $r_{P,1} - r_{P,2}$ [max. width][b] (m) | | |
| Unstable | 18 | 10–37 [12] | 8–59 [21] | 6–107 [40] | 5–206 [74] |
| Neutral | 27 | 13–75 [22] | 9–143 [45] | 7–338 [102] | 5–915 [247] |
| Stable | 35 | 15–126 [33] | 10–287 [76] | 7–897 [206] | 5–3545 [652] |
| **(2) Single cross-wind integrated ($\Pi_C$)** | | | | | |
| | $r_{max}$ (m) | | $x_P$ (m) | | |
| Unstable | 23 | 27 | 46 | 86 | 168 |
| Neutral | 39 | 56 | 112 | 269 | 734 |
| Stable | 53 | 94 | 223 | 706 | 2805 |
| **(3) Annular climatology ($\Pi^*$)** | | | | | |
| | $r_{max}$ (m) | | $r_{P,1} - r_{P,2}$ (m) | | |
| Unstable | 18 | 11–29 | 8–47 | 6–87 | 5–170 |
| Neutral | 26 | 14–57 | 10–113 | 7–272 | 5–740 |
| Stable | 34 | 16–95 | 11–225 | 7–711 | 5–2816 |

[a] Symbols: $P$ = proportion of the measured flux originating from the surface elements within the dimensions indicated; $r_{max}$ = distance of the footprint maximum; $r_{P,1}, r_{P,2}$ = distances between which the surface elements with the largest contribution to $P$ are located; $x_P$ = distance integrated from the EC tower location within which $P$ originates from.

[b] Maximum width of the source area.

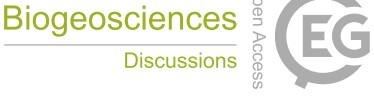



Table 4. Proportions (%) of different land cover classes as weighted by the mean footprint function during the growing season ("Weighted") and their areal coverages within the study area ("Area"), within the source area defined by the 90 % cumulative footprint ("Area, 90 %"; Eq. 3, $P$ = 90 %), within a distance of 1 km from the EC tower ("Area, 1 km") and within a 35.8 km$^2$ region ("Region", Fig. S1 in the Supplement). The values in parentheses indicate the proportions if the water pixels representing marine areas are removed and the integrated footprint within the study area is scaled to 100 %.

| Land cover class | Weighted | | Area | | Area, 90 % | | Area, 1 km | | Region | |
|---|---|---|---|---|---|---|---|---|---|---|
| Wet fen | 8.5 | (11.9) | 15.1 | (17.7) | 15.0 | (15.1) | 17.9 | (19.7) | 15.6 | (16.4) |
| Dry fen | 16.2 | (17.5) | 10.9 | (12.8) | 10.3 | (10.3) | 10.4 | (11.4) | 11.1 | (11.6) |
| Bog | 16.9 | (17.7) | 10.9 | (12.8) | 22.9 | (23.0) | 13.0 | (14.3) | 8.7 | (9.1) |
| Graminoid tundra | 11.1 | (15.1) | 5.6 | (6.6) | 11.5 | (11.6) | 6.9 | (7.6) | 3.2 | (3.4) |
| Flood meadow | 3.1 | (4.0) | 0.6 | (0.7) | 1.4 | (1.4) | 0.8 | (0.9) | 0.4 | (0.4) |
| Shrub tundra | 12.2 | (11.5) | 17.9 | (21.1) | 18.2 | (18.2) | 17.8 | (19.6) | 26.2 | (27.4) |
| Lichen tundra | 14.3 | (12.1) | 10.5 | (12.4) | 10.9 | (10.9) | 9.9 | (10.9) | 10.7 | (11.1) |
| Bare ground | 12.4 | (9.9) | 11.6 | (13.6) | 8.0 | (8.0) | 12.8 | (14.1) | 14.6 | (15.3) |
| Water | 1.0 | (0.3) | 16.9 | (2.3) | 1.9 | (1.4) | 10.6 | (1.6) | 9.4 | (5.3) |
| Out[a] | 4.3 | (0.0) | – | – | – | – | – | – | – | – |
| Total | 100.0 | (100.0) | 100.0 | (100.0) | 100.0 | (100.0) | 100.0 | (100.0) | 100.0 | (100.0) |

[a] Cumulative proportion of the footprint exceeding the limits of the study area

Table 5. Estimated fluxes for the aggregated land cover classes.

| LCC group description | CH$_4$ flux (µg m$^{-2}$ s$^{-1}$) | 95 % confidence interval (µg m$^{-2}$ s$^{-1}$) |
|---|---|---|
| Strong source | 0.949 | [0.871, 1.028] |
| Moderate source | 0.264 | [0.180, 0.348] |
| Sink | –0.131 | [–0.172, –0.089] |
| Neutral | –0.007 | [–0.035, 0.021] |





Table 6. Upscaling of $CH_4$ fluxes ($\mu g\ m^{-2}\ s^{-1}$) for different areas based on the LCC group-specific flux data shown in Table 5.

| LCC group | Study area[a] (6.3 km$^2$) | | Reduced area[a] (3.1 km$^2$) | | Region[a] (35.8 km$^2$) | |
|---|---|---|---|---|---|---|
| | Coverage[b] (%) | Flux[c] | Coverage (%) | Flux | Coverage (%) | Flux |
| Strong source | 17.7 | 0.168 | 20.2 | 0.191 | 15.1 | 0.144 |
| | | (91.8 %) | | (94.8 %) | | (89.8 %) |
| Moderate source | 19.5 | 0.052 | 17.4 | 0.046 | 20.3 | 0.054 |
| | | (28.3 %) | | (22.7 %) | | (33.5 %) |
| Sink | 26.0 | –0.034 | 25.0 | –0.033 | 26.4 | –0.035 |
| | | (–18.6 %) | | (–16.2 %) | | (–21.6 %) |
| Neutral | 36.8 | –0.003 | 37.5 | –0.003 | 38.1 | –0.003 |
| | | (–1.4 %) | | (–1.3 %) | | (–1.7 %) |
| Upscaled flux[d] | 0.183 | | 0.202 | | 0.160 | |
| | [0.156, 0.209] | | [0.176, 0.228] | | [0.134, 0.186] | |

[a] "Study area" refers to the circle with a radius of 1.4 km centred at the EC mast, while "Reduced area" has a 1.0 km radius; "Region" is shown in Fig. S1 in the Supplement. [b] Marine areas are excluded from upscaling. [c] Calculated as LCC group-specific flux × relative coverage. The value in parentheses equals this flux divided by the upscaled flux. [d] The values in square brackets indicate the 95 % confidence interval.




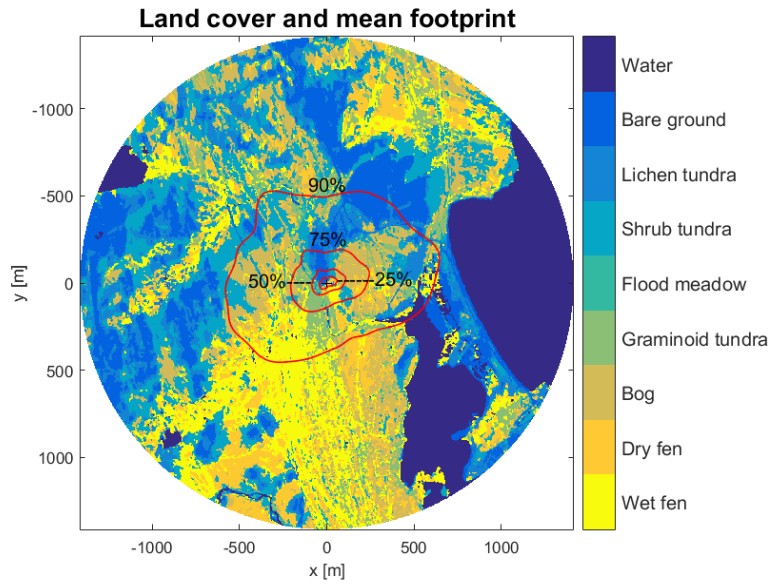

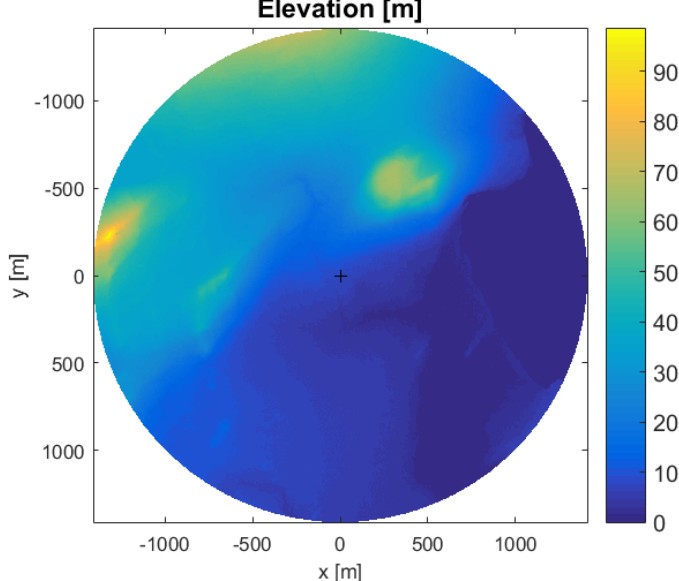



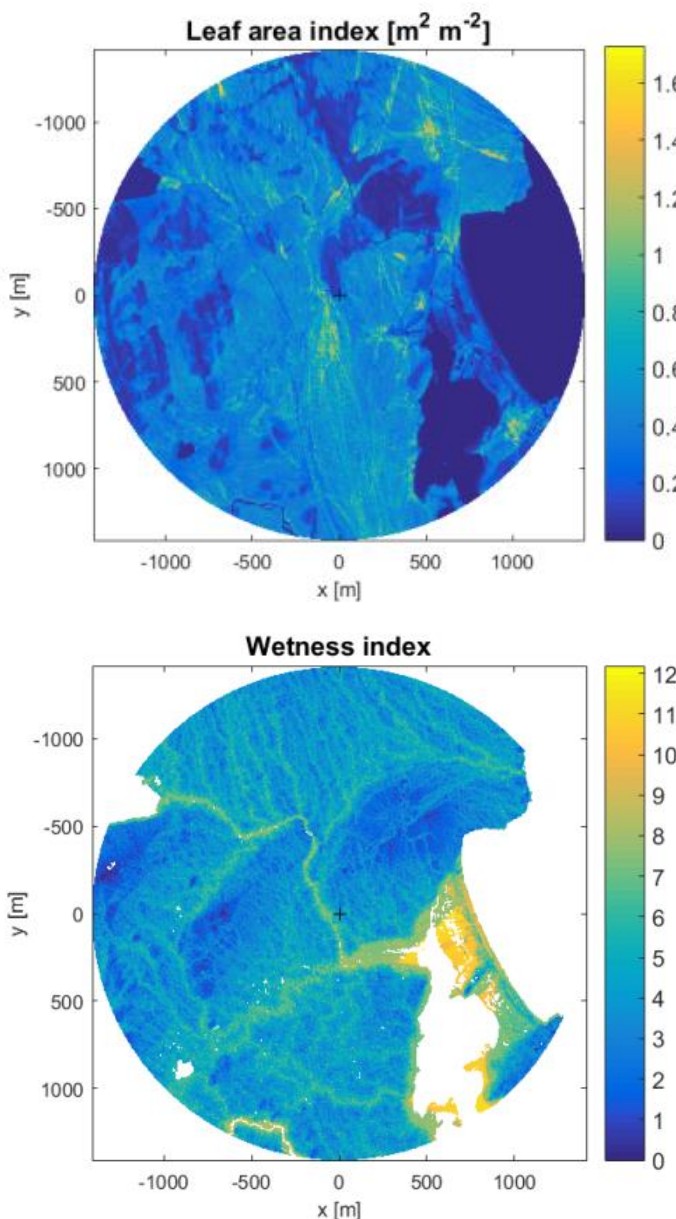

Figure 1. Land cover classes and the mean cumulative footprint during the growing season of 2014 (a), maximum leaf area index (on 12 August 2012) (b), terrain elevation (c) and topographic wetness index for terrestrial surfaces (d). The isophlets in (a) indicate the areas with a 25, 50, 75 and 90 % contribution to the measured flux. The plus sign indicates the location of the EC tower.





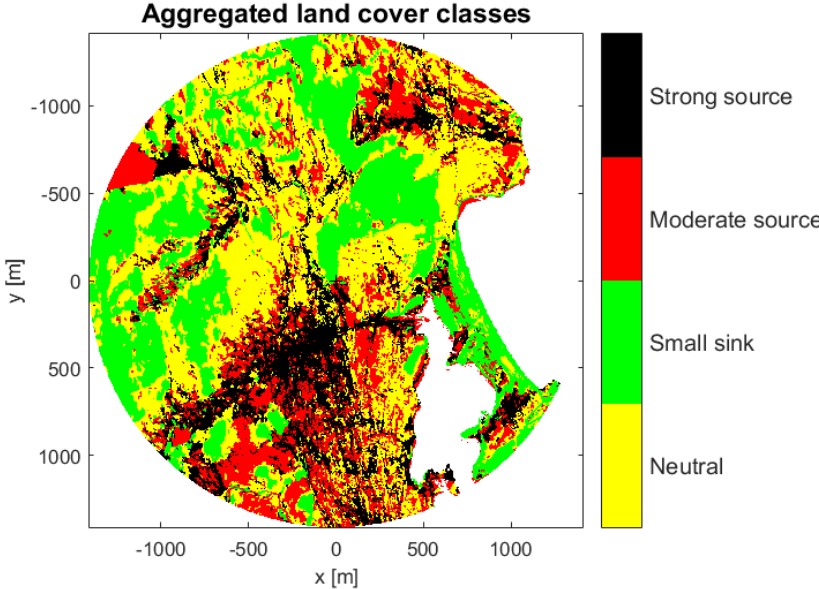

Figure 2. Distribution of the aggregated land cover classes (excluding marine areas).





5  Figure 3. Proportion of different land cover classes in the flux footprint as a function of wind direction for the three flow condition cases specified in Table 1. The rightmost panel shows the relative coverage of these classes within the study area (%).





Figure 4. The footprint-weighted and areally averaged leaf area index (a), terrain elevation (b) and topographic wetness index (c), and the corresponding sensor location biases (d–f) as a function of wind direction for the three flow condition cases specified in Table 1.




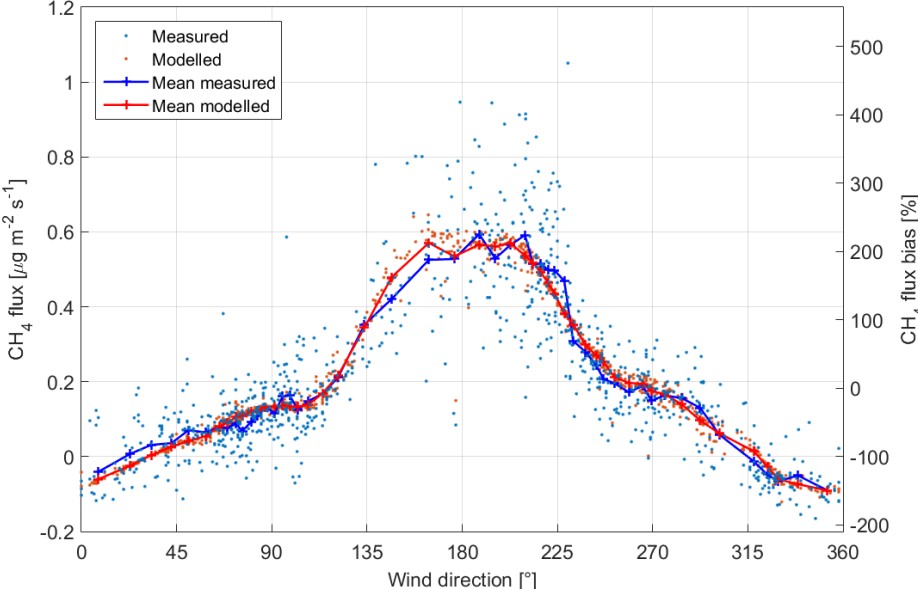

Figure 5. Measured and modelled $CH_4$ fluxes as a function of wind direction (left axis). The averaged data were calculated in 50 direction classes. The right axis indicates the sensor location bias of the measured data shown (both individual points and the mean) with respect to the mean upscaled flux within the study area (0.183 μg m$^{-2}$ s$^{-1}$).

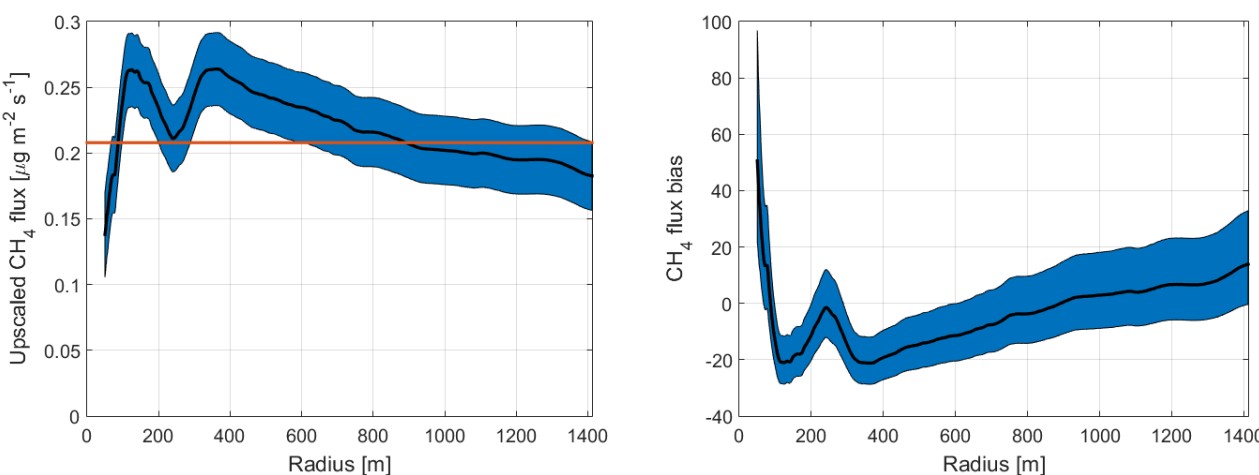

Figure 6. Upscaled $CH_4$ flux within a circular area as a function of the distance from the EC tower (a), and the corresponding sensor location bias according to Eq. (11) (b). The red line indicates the mean measured flux. The shaded area represents the

10   95 % confidence interval determined from the data shown in Table 5.