# Peer review of "Interpreting eddy covariance data from heterogeneous Siberian tundra: land cover-specific methane fluxes and spatial representativeness"

_Biogeosciences, 2018_

## Referee Comment (RC1) · Anonymous Referee #1 · 3 May 2018

Review on the manuscript titled 'Interpreting eddy covariance data from heterogeneous Siberian tundra: land cover-specific methane fluxes and spatial representativeness', submitted for publication in Biogeosciences by J.-P. Tuovinen et al. in April 2018.

The authors present a study that aims at decomposing the flux signals captured by an eddy-covariance flux system into flux signatures for individual land cover classes (LCCs) within the heterogeneous terrain surrounding the tower. In a first step, the landscape is mapped at highest resolution based on remote sensing datasets supported by ground-trothing, yielding gridded maps of LCC, elevation, NDVI and wetness. Based

on this information, the surface is subsequently categorised into 4 major groups, which are expected to feature significantly different CH4 flux signatures. Using about 8 weeks of flux data from a Siberian tundra site, Tiksi, in combination with an analytic footprint model, the authors then derive mean CH4 flux rates for each LCC group. Finally, the study evaluates the larger-scale representativeness of eddy flux measurements at the Tiksi site using the so-called sensor location bias as a metric, which compares the fractions of LCC groups within the tower footprint to those within the larger region. Here, in spite of significantly different flux signatures between groups, they find no systematic biases between study regions of different sizes.

The scientific objective behind this study is certainly relevant to the eddy-covariance (EC) community, and to all those using eddy-covariance datasets. While the EC-technique features many advantages, such as non-destructive, continuous measurements at high temporal resolution, the interpretation of results is often hampered by the fact that EC systems integrate signals from larger, often heterogeneous areas. Particularly in case of highly variable flux signatures within the landscape of interest, it is moreover often unclear how well the data from the chosen tower position represents the characteristics of the larger area. Therefore, an approach to decompose this integrated signal into separate flux signatures that represent major surface types within the field of view of the sensor could open new possibilities to data users, e.g. for modelers who this way could calibrate their frameworks for individual land cover types. Unfortunately, the presented approach falls short of this ambitious goal. The results are compromised by missing important components and over-simplifications, while the text itself is unbalanced regarding the level of detail in different sub-sections.

MAJOR COMMENTS 1.) constant flux rates: I was extremely surprised when reading the results section to find that temporal variability in flux rates is ignored, instead a constant flux rate is assigned to each of the chosen LCC groups. This constitutes an over-simplification of the tundra ecosystem in the first place, but also limits the presented approach to extremely basic overall findings. Considering that fluxes over

a period of 8 weeks from the late growing season of an Arctic ecosystem are used, besides short and mid-term climate variability the fluxes will be influenced by slowly varying conditions such as e.g. thaw depth or soil temperatures. Accordingly, it cannot be assumed that even mean flux levels for moving windows remain constant over such a long period. Even more important in this context, I would strongly assume that the different LCC groups will show a different trend in phenology in this part of the season, i.e. some may be subject to earlier senescence, and also some of them may react more strongly to environmental stress such as water limitation, or first nights with freezing temperatures. This implies that the differences/ratios between flux rates from LCC groups will not be constant over time. So what is the value of providing just a single mean flux rate per LCC group? Would the differences still be significant if short-term temporal variability, and shifts in flux rate ratios, would be taken into account? Regarding the applicability of the results, under the given circumstances the output of the flux decomposition is of no value for any other purpose than investigating a potential sensor location bias (and even here the impact is limited). To reach a broader audience, temporally varying fluxes per biome with functional links to environmental controls would have to be provided. Summarising this item, it is obvious that the chosen approach is based on a strong simplification of the actual flux patterns. This should be discussed in detail in a revised version of the manuscript. This discussion needs to be supplemented by a demonstration how variable measured net flux rates are over time, broken up into the chosen LCC groups. If it cannot be proven that the ratios between flux rates from different groups remain largely constant over time, the approach cannot be applied as is. In general, I strongly urge the authors to consider extending their approach, so temporal variability in flux rates can be considered.

2.) uncertainty estimates: The manuscript misses to even discuss some essential sources of uncertainty that influence the given approach, and those few aspects that are treated (e.g. uncertainties in maps) are only covered qualitatively. Even very easy components, such as e.g. assigning an uncertainty to the input flux rates from the EC system, which is then projected onto the modelled LCC flux rates, is missing. Most

importantly, there is no uncertainty estimate for the footprint approach. It is obvious that any source weight function can only be an approximation of the actual field of view of the sensor, as many footprint validation studies have shown in the past. In this study, however, footprint simulations are treated as a given fact. There are uncertainties in all the input parameters used to feed the footprint model, there are uncertainties associated with parameterizations/assumptions inherent to the footprint model, and there are uncertainties related to the methodology (e.g. horizontal homogeneity, stationary flow, and so on). For a modified version of this study, the authors need to provide a convincing concept to constrain the uncertainties in computed source weight functions, and how these influence the results obtained by the flux decomposition approach. In addition, the uncertainty concept should, as mentioned above, also involve the flux data uncertainty, and also the uncertainties inherent to the maps used in this study should be involved, and quantified.

3.) Validation of flux rates: In Section 3.2, the authors include a good paragraph (p.16 ll.20ff) that supports the negative flux rates found for bare soil. As part of the line of argumentation, chamber measurements from a previous study are cited. I think it's fair to assume that this study did not only measure those 32 data points cited here for bare soil, but also other components within the Tiksi landscape. Why are those not used? Having flux chamber results for the different LCC groups would be the best way to validate that the flux composition actually produced realistic results. Also, the Tiksi flux tower has been running for several years now - why restrict this study to just 8 weeks? Why not use more data, so the database is more representative, and can also resolve temporal variability? Why not split the dataset into training and validation sets, so any finding can actually be evaluated?

4.) scope of this study: With the limitations of the chosen concept (constant fluxes) as mentioned above, the authors should clearly restrict the scope of the study to an estimate to constrain sensor location bias. I do not see any other application of their method besides this (I would be glad to get convinced otherwise, e.g. by a thorough

discussion ..). I do not think they can claim to provide land cover specific CH4 flux rates, since they present one set of mean flux rates for a single period of time, nothing more. They also do not interpret EC data, since obviously there's no temporal variability, no links to environmental controls, no interpretation why certain LCC groups show different fluxes than others. What is being provided here is an extremely simplified approach to estimate flux rates per LCC group, and check if, given these flux rates, the net fluxes represent the emissions from a larger area (aka sensor location bias). Since there is also no discussion which aspects influence the performance of this approach (e.g. length scale of variability in terrain features, differences in flux rates between LCC groups, footprint variability, etc), there is no way of telling if this approach could be applied at other sites as well.

5.) a thorough discussion is simply missing! What is the implication of the findings? How could the presented approach be used? Where are the weaknesses, which factors limit the interpretation of the findings?

These are the main points of criticism. More general and technical comments are listed below.

Summarising, I think there is a lot of potential in developing solid schemes that facilitate the decomposition of eddy-covariance flux rates into signal for individual landscape components within the tower footprint. The presented approach, however, provides only a very minor first step towards this direction. The only contribution towards an improved evaluation of eddy-covariance data I currently see is that, given very strong simplifications regarding signal variability, a sensor location bias can be roughly approximated. In case such a bias is detected, however, the output of this scheme wouldn't help to overcome it. Under this light, the overall content of this paper is quite slim, particularly if many unnecessary sections providing textbook knowledge are removed. Summarising, in the present form that manuscript is far from being ready to be accepted for publication. Still, I believe the chosen topic is very relevant, and therefore I hope that the authors are willing to fix the issues raised in the comments above (and

[Figure]

below).

GENERAL COMMENTS 1.) The introduction is well structured overall, and the 3 different sets of objectives are clearly formulated. The paragraph on methane (starting p.2 l.25) is rather confusing, though, and should be revised.

2.) Section 2.2 needs a complete overhaul. Many sections, e.g. most parts of 2.2.1, are textbook knowledge, and do not need to be shown in detail herein. Section 2.2.2 is much too detailed for what actually needs to be described. You project a source weight function on gridded maps, and accumulate the weights of individual cells, sorted by categories, nothing more. Overall, this whole section is much too long. I suggest to revise it to the following structure: - 2.2.3 should be moved to the front - 2.2.2 should be shortened, and simplified, coming next - 2.2.1 should be discarded entirely - 2.2.4 should be moved as part of the results section

3.) In Section 2.3, the ordering of the information should be revised. Many pieces of information given in 2.3.2 were needed to interpret the text in 2.3.1, for example.

4.) Results Section 3.1: The first part on general footprint characteristics should be removed (P1, P2). After all, what you basically state here is the obvious fact that footprint areas grow with stable stratification. The authors may move Table 3 to the appendix, and refer to it in the main text in case any reader wants to see the details, but this is clearly not part of the main story. The center part, highlighting the heterogeneity of surface characteristics within the footprint, reads well (P3-P5). The last part (P6+) should be revised - it is informative to describe a sensor location bias using the different surface characteristics, but the current format is confusing, using too many versions of a reference area (also Table 4 should be reduced).

4.) p.16 l.10ff: I don't think it makes much sense to compare the Tiksi flux rates against values from other sites without also comparing environmental conditions, and the measurement approaches.

5.) results section 3.3: It is confusing, and actually not understandable, why so many different reference areas have been used to compare the footprint LCC composition to. This actually leaves the impression that the authors were searching for a nice configuration that can demonstrate that the EC measurements are actually well representative (e.g. p.17 l.27 ' the sensor location bias could be minimized by reducing the radius to 800– 1000 m'). What is the value in such an exercise? People who are interested in using EC data want to know how well they represent a LARGE area.

TECHNICAL COMMENTS p.3 l. 16: I don't see a connection between spatial heterogeneity and the need for long-term measurements . . .??

Section 2.1.1: A bit too brief. Soil types could be mentioned, and it should be mentioned that vegetation is given in a different subsection.

Section 2.1.2: The outline of the QC is too short. What exactly was done regarding instationarity, for example? How were unphysical outliers defined? And how were the gaps treated in the end?

Section 2.1.3: I suggest moving the definition of PCTs into a table. It should be mentioned that the dominant vegetation, and other characteristics, are described later in 2.1.4

p.6 l.16: The authors should decide if they want to use LCC or PCT as a term for this classification. Using both is very confusing!

Fig.4: the lower 3 panels are not necessary , since they show the same patterns as above, only normalized against the black dashed line

---

## Referee Comment (RC2) · Anonymous Referee #2 · 5 Jul 2018

The manuscript by Tuovinen et al. aims to use a detailed footprint analysis of a flux tower in Northeast Siberia to identify sensor location bias, while making use of high resolution satellite imagery. I think this study is interesting but, like the other referee, I think this paper could be a lot more effective. I agree with much of what is said in the other review, but I have a few additional comments.

First of all: if sensor location bias is your main goal, this can be done a lot easier through an analysis of the land cover classification from the satellite image. The heterogeneity of the landscape is already captured in there, and by taking the average of a

hypothetical footprint area at random points within the satellite image, it can be clarified which locations resemble the larger area the most. A rough estimate of the footprint would be required but this would be a lot simpler than the approach in the paper.

As for the footprint area itself: the choice of the Korman and Meixner model is curious. There have been many advances in footprint analysis since, see for example Kljun et al (2015) who described a two-dimensional footprint model that already gives the footprint contribution for each part of the footprint. The model is freely available at http://footprint.kljun.net for Matlab, R and python.

If the authors had used this model, they could have simplified a large part of the methods, which would really help with the readability of the paper. At present, the long list of equations makes it hard to follow what the direction of the paper is, especially since many of them are not referred to later on. Like the author referee said, we don't need textbook knowledge. Sections 2.1 and 2.2 should be drastically shortened to only those equations directly relevant to the paper.

Also, I don't understand why there are not more flux chamber measurements in the area, but only from bare soil. The spatial heterogeneity of methane fluxes can be much better identified from direct measurements rather than the inverse method presented here. Please, when reporting EC methane flux measurements, use nmol/m2/s rather than micrograms.

Finally, the authors should put their research in a better context compared to existing literature. The discussion is very limited, and previous studies that have had detailed footprint analyses or emphasized the spatial variation of methane fluxes are either ignored or referenced too briefly. See for example Matthes et al (2014) and Maruschak et al (2016) for two prime examples, but also the paper by Parmentier et al. (2011) which you do cite but without mentioning that it also dealt with methane flux heterogeneity. There are many more, and this needs to be recognized.

Other comments:

[Figure]

Page 2, line 27: it would make more sense to mention only natural emissions, since this paper focuses on that. This number of 560 Tg/yr is anthropogenic plus natural.

Page 3, line 13-15: this has been known for decades, and should not be presented as something new. See for example Christensen, 1993; Torn and Chapin, 1993; Whiting and Chanton, 1992.

Page 5, line 27-28: a table with vegetation descriptions for each class would be helpful.

Page 6, line 9: have you considered NDWI as another wetness index?

Page 10, line 23: is this normalization necessary? This paragraph seems like a complex way of simply saying that you have some unknown data in your average. In any case, at 1.4 km from the tower the effect would be negligible.

Page 12, line 23: which Eriophorum species? Not all of them are high emitters of methane, like Eriophorum vaginatum.

Page 15, line 6-7: "the areal coverage of the LCC within the study area"? what do you mean? The LCC map?

Page 17, line 27: this feels like cheating. The tower is not representative for the larger region, so you reduce the region. Please don't.

Page 18, line 2: there's no doubt that lakes are a significant source of methane in the Arctic. Please rewrite this sentence to remove the suggestion that it may not be (e.g. 'Nevertheless, Arctic lakes and ponds emit significant amounts of CH4 in general).

Page 31, figure 1. Please don't use a continuous color map for land cover. Use discrete colors.

References

Christensen, T. R.: Methane emission from Arctic tundra, Biogeochemistry, 21(2), 117–139, 1993.

[Figure]

Kljun, N., Calanca, P., Rotach, M. W. and Schmid, H. P.: A simple two-dimensional parameterisation for Flux Footprint Prediction (FFP), Geosci. Model Dev., 8(11), 3695–3713, 2015.

Marushchak, M. E., Friborg, T., Biasi, C., Herbst, M., Johansson, T., Kiepe, I., Liimatainen, M., Lind, S. E., Martikainen, P. J., Virtanen, T., Soegaard, H. and Shurpali, N. J.: Methane dynamics in the subarctic tundra: combining stable isotope analyses, plot- and ecosystem-scale flux measurements, Biogeosciences, 13(2), 597–608, 2016.

Matthes, J. H., Sturtevant, C., Verfaillie, J., Knox, S. and Baldocchi, D.: Parsing the variability in CH4 flux at a spatially heterogeneous wetland: Integrating multiple eddy covariance towers with high‐resolution flux footprint analysis, J. Geophys. Res. Biogeosci., 119(7), 1322–1339, 2014.

Torn, M. S. and Chapin, F. S., III: Environmental and biotic controls over methane flux from arctic tundra, Chemosphere, 26(1-4), 357–368, 1993.

Whiting, G. J. and Chanton, J. P.: Plant‐dependent CH4 emission in a subarctic Canadian fen, Glob. Biogeochem Cycles, 6(3), 225–231, doi:10.1029/92GB00710, 1992.

---

## Author Comment (AC1) · 20 Jul 2018

We thank the referee for the extensive criticism. We respond to each comment in the following.

First, we should mention that our original plan was to submit another, more data-oriented manuscript at the same time. This paper would present longer-term data and its processing in more detail and constitute a more conventional study of flux variations and their environmental controls. Unfortunately, the preparation of that manuscript was

delayed and thus a concurrent submission was not possible. As we did not find it appropriate to refer to a "manuscript in preparation", we removed such citations at a final stage. Thus it is possible that some essential information is missing and we need to complement the manuscript with material that supports our arguments and presentation.

MAJOR COMMENTS

Comment:

1.) constant flux rates: I was extremely surprised when read- ing the results section to find that temporal variability in flux rates is ignored, instead a constant flux rate is assigned to each of the chosen LCC groups. This constitutes an over-simplification of the tundra ecosystem in the first place, but also limits the presented approach to extremely basic overall findings. Considering that fluxes over a period of 8 weeks from the late growing season of an Arctic ecosystem are used, besides short and mid-term climate variability the fluxes will be influenced by slowly varying conditions such as e.g. thaw depth or soil temperatures. Accordingly, it cannot be assumed that even mean flux levels for moving windows remain constant over such a long period. Even more important in this context, I would strongly assume that the different LCC groups will show a different trend in phenology in this part of the sea- son, i.e. some may be subject to earlier senescence, and also some of them may react more strongly to environmental stress such as water limitation, or first nights with freez- ing temperatures. This implies that the differences/ratios between flux rates from LCC groups will not be constant over time. So what is the value of providing just a single mean flux rate per LCC group? Would the differences still be significant if short-term temporal variability, and shifts in flux rate ratios, would be taken into account? Regard- ing the applicability of the results, under the given circumstances the output of the flux decomposition is of no value for any other purpose than investigating a potential sensor location bias (and even here the impact is limited). To reach a broader audience, tem- porally vary- ing fluxes per biome with functional links to environmental controls would have to be

provided. Summarising this item, it is obvious that the chosen approach is based on a strong simplification of the actual flux patterns. This should be discussed in detail in a revised version of the manuscript. This discussion needs to be supple- mented by a demonstration how variable measured net flux rates are over time, broken up into the chosen LCC groups. If it cannot be proven that the ratios between flux rates from different groups remain largely constant over time, the approach cannot be ap- plied as is. In general, I strongly urge the authors to consider extending their approach, so temporal variability in flux rates can be considered.

Reply:

While it is plausible to assume that there should be significant temporal flux variabil- ity driven by environmental controls, such as soil temperature, there is obviously no guarantee that these would constitute the main factor controlling the variations in the observed fluxes. At Tiksi, with a heterogeneous landscape, this variation during the study period is overshadowed by variations in the flux footprint, i.e. the varying contri- bution of different land cover types. The following arguments support this conclusion: (1) There is no significant correlation between the measured CH4 flux and either soil temperature or air temperature. There is no such correlation either if the data are lim- ited to land cover class (LCC) group-specific cases in which a LCC group dominates. (2) Figure 5 shows that the observed CH4 flux depends strongly on wind direction. This dependency reflects the variability of flux footprint rather than environmental fac- tors. The variance of measured 30-min fluxes is 0.049 microg2/m4/s2 . Calculated from the binned mean fluxes (50 classes in Fig. 5), the variance related to the direc- tional pattern is 0.040 microg2/m4/s2, i.e. 82% of the total variance. (3) The regression model presented shows that LCC proportions explain 80% of the flux variation (p.15, l.32). We conclude that these results indicate that the short-term variability in the ob- served fluxes is predominantly due to footprint variability and any environmental control can have a secondary effect only.

This does not rule out the role of phenology and longer-term trends in soil temperature, for example, in affecting the longer-term variations of CH4 fluxes. In the revised manuscript, we will analyse the data also on a weekly basis. This analysis will indicate that the differences between the LCC group-specific fluxes persist systematically but also that the fluxes indeed vary even within the rather limited study period; in the revised manuscript we will show how this relates to the trends in soil temperature and LAI. The weekly mean fluxes will be used for upscaling, too, and we can show that our conclusion about the spatial representativeness of the EC measurements is robust. The temporally resolved analysis also provides material for an enhanced discussion of the results, for example indicating that no statistically significant estimates can be obtained when the data set becomes too limited.

Comment:

2.) uncertainty estimates: The manuscript misses to even discuss some essential sources of uncertainty that influence the given approach, and those few aspects that are treated (e.g. uncertainties in maps) are only covered qualitatively. Even very easy components, such as e.g. assigning an uncertainty to the input flux rates from the EC system, which is then projected onto the modelled LCC flux rates, is missing. Most importantly, there is no uncertainty estimate for the footprint approach. It is obvious that any source weight function can only be an approximation of the actual field of view of the sensor, as many footprint validation studies have shown in the past. In this study, however, footprint simulations are treated as a given fact. There are uncertainties in all the input parameters used to feed the footprint model, there are uncertainties associated with parameterizations/assumptions inherent to the footprint model, and there are uncertainties related to the methodology (e.g. horizontal homogeneity, stationary flow, and so on). For a modified version of this study, the authors need to provide a convincing concept to constrain the uncertainties in computed source weight functions, and how these influence the results obtained by the flux decomposition approach. In addition, the uncertainty concept should, as mentioned above, also involve the flux data uncertainty, and also the uncertainties inherent to the maps used in this study should

be involved, and quantified.

Reply:

It is incorrect to state that we do not address or quantify uncertainties. The LCC group-specific flux estimates are presented with the 95% confidence intervals (Table 5), which are based on a heteroskedasticity and autocorrelation consistent estimator (Sect. 2.3.1). These error estimates are used for upscaling the fluxes, the results of which are also shown with quantitative uncertainty estimates (Table 6 and Figure 6). We also report the LCC classification accuracies (Sect. 2.1.3) and discuss the related uncertainty (Sect. 3.2, end). As explained in Sect. 2.3.1, we assume that the confidence intervals represent the integrated effect of different error sources, including the measurement data, LCC classification and footprint modelling. We believe that this is a more appropriate method than a bottom-up approach, in which individual error estimates should be first allocated to each error source assumed and then propagated to estimate the total uncertainty. Whereas it is obvious that footprint modelling has uncertainties, both structural and input related, it is not obvious at all how this could be quantified for a meaningful error estimate. However, we agree that the uncertainties related to footprints should be made more explicit. We will add discussion that explains that the footprint dimensions and representativeness metrics presented in the manuscript are affected by model uncertainties, including citations to footprint validation studies.

Comment:

3.) Validation of flux rates: In Section 3.2, the authors include a good paragraph (p.16 ll.20ff) that supports the negative flux rates found for bare soil. As part of the line of argumentation, chamber measurements from a previous study are cited. I think it's fair to assume that this study did not only measure those 32 data points cited here for bare soil, but also other components within the Tiksi landscape. Why are those not used? Having flux chamber results for the different LCC groups would be the best way to

validate that the flux composition actually produced realistic results. Also, the Tiksi flux tower has been running for several years now - why restrict this study to just 8 weeks? Why not use more data, so the database is more representative, and can also resolve temporal variability? Why not split the dataset into training and validation sets, so any finding can actually be evaluated?

Reply:

The reviewer is right that the chamber measurements cited in the manuscript were not limited to bare soil. We also agree that such data would be useful for validating the present results. The reason for not using the chamber data more extensively is that the experimental design was incomplete: the number of chamber plots was modest, and the reach from the EC mast was limited due to the use of an online gas analyser. Moreover, the chamber plots did not fully correspond to the land cover classification that was subsequently developed and used in the present study. Thus it is not possible to use these data for a proper validation of the flux decomposition. The bare soil data were introduced to provide support for the surprisingly high uptakes rates observed. However, we can report here that the overall pattern the chamber data depicts is consistent: wet fens appear as strong CH4 emitters (two plots, 32 observations at each plot, means of 0.56 and 3.8 microg/m2/s) and dry fens as moderate emitters (four plots, 31/32 observations, mean 0.25 microg/m2/s).

It is true that the Tiksi tower has been running for several years now. As indicated above, a paper analysing longer-term data is in progress, while the aim of the present manuscript is different and achievable with data from a more limited period representing a well-defined season.

As for training/validation, we did validate the results by splitting the dataset into training and validation sets. The approach is described in Sect 2.3.1 (end). The validation statistics show a good performance against independent data (p.16, l.6-8).

Comment:

4.) scope of this study: With the limitations of the chosen concept (constant fluxes) as mentioned above, the authors should clearly restrict the scope of the study to an estimate to constrain sensor location bias. I do not see any other application of their method besides this (I would be glad to get convinced otherwise, e.g. by a thorough discussion ..). I do not think they can claim to provide land cover specific CH4 flux rates, since they present one set of mean flux rates for a single period of time, nothing more. They also do not interpret EC data, since obviously there's no temporal variability, no links to environmental controls, no interpretation why certain LCC groups show different fluxes than others. What is being provided here is an extremely simplified approach to estimate flux rates per LCC group, and check if, given these flux rates, the net fluxes represent the emissions from a larger area (aka sensor location bias). Since there is also no discussion which aspects influence the performance of this approach (e.g. length scale of variability in terrain features, differences in flux rates between LCC groups, footprint variability, etc), there is no way of telling if this approach could be applied at other sites as well.

Reply:

Unfortunately, we fail to see the logic of the comment that we cannot "claim to provide land cover specific CH4 flux rates, since [we] present one set of mean flux rates for a single period of time, nothing more." It is true that we presented average fluxes, but any averaging is irrelevant to the question whether we provided (statistically significant estimates of) land cover specific fluxes. We obviously did. However, in the revised version we will address temporal variation by analysing the data in shorter periods.

Similarly, the comment "[t]hey also do not interpret EC data, since obviously there's no temporal variability, no links to environmental controls, no interpretation why certain LCC groups show different fluxes than others" seems odd. The meaning of "interpretation" in the context of this manuscript should be clear already from the latter part of the title and the objectives listed in the introduction, the latter of which the reviewer commends later in the review: "the 3 different sets of objectives are clearly formulated". These objectives do not include temporal variability or environmental controls; in contrast, they explicitly refer to mean fluxes (p.4, l.25). Please note also that in conclusions (p.19, l.9-10) we state that the environmental controls and long-term data will be studied in a follow-up paper. We realise this is possible and even necessary for the present manuscript, and the temporally resolved analysis that will be included will address these aspects.

As for the last alleged omission, i.e. no discussion about the different fluxes among the LCC groups, this is formulated, on the basis of a literature survey, as a statistical hypothesis. This is explained in Sect. 2.3.2. The success of the regression model, in terms of both statistical significance and the logic of results, confirms this hypothesis and consequently provides credence to the assumptions behind the flux differences. This is discussed in Sect. 3.2, in which we report previously measured fluxes for different tundra surfaces. We will complement this discussion by outlining mechanisms that are known to explain the differences.

Comment:

5.) a thorough discussion is simply missing! What is the implication of the findings? How could the presented approach be used? Where are the weaknesses, which factors limit the interpretation of the findings?

Reply:

While some implications of the study are presented in the conclusions ("An important implication emerging from our results...", p.18,l.26), we agree that the discussion related to the applicability of the approach presented is too limited. We will add discussion about the feasibility of the statistical model, drawing upon the new analysis of temporal variations to be included.

GENERAL COMMENTS

Comment:

1.) The introduction is well structured overall, and the 3 differ- ent sets of objectives are clearly formulated. The paragraph on methane (starting p.2 l.25) is rather confusing, though, and should be revised.

Reply:

We will revise this paragraph.

Comment:

2.) Section 2.2 needs a complete overhaul. Many sections, e.g. most parts of 2.2.1, are textbook knowledge, and do not need to be shown in detail herein. Section 2.2.2 is much too detailed for what actually needs to be described. You project a source weight function on gridded maps, and accumulate the weights of individual cells, sorted by categories, nothing more. Overall, this whole section is much too long. I suggest to revise it to the following structure: - 2.2.3 should be moved to the front - 2.2.2 should be shortened, and simplified, coming next - 2.2.1 should be discarded entirely - 2.2.4 should be moved as part of the results section

Reply:

Our idea was to show that the source area can be defined in different ways, and our results show that it is important to present an exact definition when reporting footprint dimensions or referring to the study area. There are numerous papers in which the footprint concept is used very loosely, even in a misleading way, but we refrained from specifying them in the text. However, to improve the focus of the manuscript, we will remove Sect. 2.2.1 and include a shortened version as an appendix.

We do not understand why Sect. 2.2.2, which only covers less than 1.5 manuscript pages, would be "much too detailed". It provides the mathematical definition of the variables we use in our analysis – "nothing more", to cite the reviewer, and we do not claim otherwise in the manuscript. We feel that an exact description of the methods should be encouraged rather than discouraged. Mathematical formulation facilitates

such exactness (and someone so inclined may even find it useful to see the EC-related averaging process formalised as in Eq. 8, for example).

Comment:

3.) In Section 2.3, the ordering of the information should be revised. Many pieces of information given in 2.3.2 were needed to interpret the text in 2.3.1, for example

Reply:

We will check the ordering of information in this section and revise the text accordingly.

Comment:

4.) Results Section 3.1: The first part on general footprint characteristics should be removed (P1, P2). After all, what you basically state here is the obvious fact that footprint areas grow with stable stratification. The authors may move Table 3 to the appendix, and refer to it in the main text in case any reader wants to see the details, but this is clearly not part of the main story. The center part, highlighting the heterogeneity of surface characteristics within the footprint, reads well (P3-P5). The last part (P6+) should be revised - it is informative to describe a sensor location bias using the different surface characteristics, but the current format is confusing, using too many versions of a reference area (also Table 4 should be reduced).

Reply:

While we agree that it is well-known that the footprint area increases with atmospheric stability, this is not presented as a conclusion of the present study; we state that the results show "expected qualitative features" (p.13, l.11). Rather, we present quantitative dimensions of the flux footprint for this site, which we believe constitute information that is essential for further studies that use the EC data from this site. Furthermore, we compare different source area definitions, showing substantial differences, and conclude that it is important to report an exact definition (see above). However, we agree that this may appear a side track and will move Table 3 to an appendix and modify the

first paragraphs of Sect. 3.1. We will also reduce the number of reference areas and simplify Table 4 and the related text in Sect 3.1. accordingly.

Comment:

4.) p.16 l.10ff: I don't think it makes much sense to compare the Tiksi flux rates against values from other sites without also comparing environmental conditions, and the measurement approaches

Reply:

We think it is a common procedure to compare new data to previous results, even if there may be differences in site characteristics and environmental conditions. We compare the LCC group-specific fluxes with an extensive set of chamber-based measurements (cf. Major comment 3) during the growing season (as detailed on p.16, l.10-13), indicating that our LCC group-specific flux estimates are reasonable. We also present corresponding EC results from comparable sites and observe that the mean $CH_4$ flux at Tiksi is within the variation in the mean summer flux among these sites.

Comment:

5.) results section 3.3: It is confusing, and actually not understandable, why so many different reference areas have been used to compare the footprint LCC composition to. This actually leaves the impression that the authors were searching for a nice configuration that can demonstrate that the EC measurements are actually well representative (e.g. p.17 l.27 ' the sensor location bias could be minimized by reducing the radius to 800– 1000 m'). What is the value in such an exercise? People who are interested in using EC data want to know how well they represent a LARGE area.

Reply:

In practice, an EC study site is defined as a more or less arbitrary area surrounding the EC mast. In our case, we selected this area according to a seemingly objective criterion based on a footprint dimension (95% coverage in neutral conditions), but obviously we

could have chosen a different target area. In any case, the spatial representativeness of the EC data collected depends on the corresponding footprint climatology. Our analysis shows that our CH4 flux measurements are representative of the original reference area, given the uncertainty range of the upscaled flux. However, the mean sensor bias would be smaller for a smaller target area, which would be as logical a choice as the original one. It is important to know how well the EC data represent a large area, which we also assess, but first it is important to understand what is the surface configuration that you actually are measuring. We will reduce the number of different reference areas and try to clarify this discussion in the revised version.

TECHNICAL COMMENTS

Comment:

p.3 l. 16: I don't see a connection between spatial hetero- geneity and the need for long-term measurements ...??

Reply:

The idea was to imply that for representative sampling more data are needed for heterogeneous than homogeneous surfaces. We will remove 'long-term'.

Comment:

Section 2.1.1: A bit too brief. Soil types could be mentioned, and it should be mentioned that vegetation is given in a different subsection. Section 2.1.2: The outline of the QC is too short. What exactly was done regarding instationarity, for example? How were unphysical outliers defined? And how were the gaps treated in the end?

Reply:

We agree that these sections are too brief. This relates to the delayed paper mentioned above, which would have provided more details. We will add more information about the soil types and data processing.

Comment:

Section 2.1.3: I suggest moving the definition of PCTs into a table. It should be mentioned that the dominant vegetation, and other characteristics, are described later in 2.1.4

Reply:

This may introduce some repetition, but we will prepare a table summarising the LCC properties.

Comment:

p.6 l.16: The authors should decide if they want to use LCC or PCT as a term for this classification. Using both is very confusing!

Reply:

Plant functional types (PFTs) and land cover classes (LCCs) do not refer to the same thing. The vegetation was surveyed as PFTs, and each of the LCCs may contain several PFTs. The confusion probably stems from the inexact formulation on p.5. (l.26-), which aims to list the LCCs rather than the PFTs. We will rephrase this sentence.

Comment:

Fig.4: the lower 3 panels are not necessary , since they show the same patterns as above, only normalized against the black dashed line

Reply:

The reviewer is right: the patterns in the lower panels are the same as in the upper ones. We will remove the lower panels and add new right axes for bias (%) to the upper ones.

---

## Author Comment (AC2) · 20 Jul 2018

We thank the referee for the comments. We respond to each of them in the following.

Comment:

The manuscript by Tuovinen et al. aims to use a detailed footprint analysis of a flux tower in Northeast Siberia to identify sensor location bias, while making use of high resolution satellite imagery. I think this study is interesting but, like the other referee, I think this paper could be a lot more effective. I agree with much of what is said in the

other review, but I have a few additional comments.

Reply:

Please see also our response to Referee 1.

Comment:

First of all: if sensor location bias is your main goal, this can be done a lot easier through an analysis of the land cover classification from the satellite image. The heterogeneity of the landscape is already captured in there, and by taking the average of a hypothetical footprint area at random points within the satellite image, it can be clarified which locations resemble the larger area the most. A rough estimate of the footprint would be required but this would be a lot simpler than the approach in the paper.

Reply:

Estimation of the sensor location bias is included in one of the three main aims of the manuscript; these aims are reported in the Introduction (p.4, l.18-). We agree that land cover maps provide useful information that can guide the selection of an EC site location. However, the actual representativeness of EC measurements against an areal average cannot be evaluated without considering the unequal weighting of the surface elements in each measurement.

Comment:

As for the footprint area itself: the choice of the Korman and Meixner model is curious. There have been many advances in footprint analysis since, see for example Kljun et al (2015) who described a two-dimensional footprint model that already gives the footprint contribution for each part of the footprint. The model is freely available at http://footprint.kljun.net for Matlab, R and python.

Reply:

Despite recent advances in footprint modelling, the Korman and Meixner model is still

commonly used since it is based on a valid micrometeorological theory of surface layer turbulence and involves a minimum number of additional assumptions for practicality. It is especially suitable for smooth surfaces, such as tundra. Furthermore, all the input data are directly obtained from the EC measurements, whereas an estimate of the mixing height is required if the Kljun et al. model is used. We will add discussion about the uncertainties related to footprint modelling.

Comment:

If the authors had used this model, they could have simplified a large part of the methods, which would really help with the readability of the paper. At present, the long list of equations makes it hard to follow what the direction of the paper is, especially since many of them are not referred to later on. Like the author referee said, we don't need textbook knowledge. Sections 2.1 and 2.2 should be drastically shortened to only those equations directly relevant to the paper.

Reply:

The choice of the footprint model employed has nothing to do with the equations presented in the manuscript. The equations in Sections 2.2.1 and 2.2.2 are totally independent of the Korman and Meixner model and would be exactly the same for any other model. Using the Kljun et al. model would not simplify the methods a bit, nor would it make any practical difference in terms of computational complexity. Both models provide the footprint field (variable f in the text) for each 30-min averaging period, which data are then processed as described in the manuscript. As for the equations, two of them are not referred to later: Eq. (6) was included for completeness and can be removed, and Eq. (12) is an essential part of the methods. To improve the focus of the manuscript, we will remove Sect. 2.2.1 and include a shortened version, together with the related results, as an appendix.

Comment:

[Figure]

Also, I don't understand why there are not more flux chamber measurements in the area, but only from bare soil. The spatial heterogeneity of methane fluxes can be much better identified from direct measurements rather than the inverse method presented here.

Reply:

Essentially the same question was raised by Referee 1, so we repeat our reply: The reviewer is right that the chamber measurements cited in the manuscript were not limited to bare soil. We also agree that such data would be useful for validating the present results. The reason for not using the chamber data more extensively is that the experimental design was incomplete: the number of chamber plots was modest, and the reach from the EC mast was limited due to the use of an online gas analyser. Moreover, the chamber plots did not fully correspond to the land cover classification that was subsequently developed and used in the present study. Thus it is not possible to use these data for a proper validation of the flux decomposition. The bare soil data were introduced to provide support for the surprisingly high uptakes rates observed. However, we can report here that the overall pattern the chamber data depicts is consistent: wet fens appear as strong CH4 emitters (two plots, 32 observations at each plot, means of 0.56 and 3.8 microg/m2/s) and dry fens as moderate emitters (four plots, 31/32 observations, mean 0.25 microg/m2/s).

Comment:

Please, when reporting EC methane flux measurements, use nmol/m2/s rather than micrograms.

Reply:

(kilo)gram is an SI unit, so microgram/m2/s is a valid unit for mass flux density.

Comment:

Finally, the authors should put their research in a better context compared to existing

literature. The discussion is very limited, and previous studies that have had detailed footprint analyses or emphasized the spatial variation of methane fluxes are either ignored or referenced too briefly. See for example Matthes et al (2014) and Maruschak et al (2016) for two prime examples, but also the paper by Parmentier et al. (2011) which you do cite but without mentioning that it also dealt with methane flux heterogeneity. There are many more, and this needs to be recognized.

Reply:

Quite a few papers dealing with footprint analysis and methane fluxes are mentioned in the Introduction and Sect. 3.2, but we will enhance the discussion in this respect and include the suggested references.

Other comments:

Comment:

Page 2, line 27: it would make more sense to mention only natural emissions, since this paper focuses on that. This number of 560 Tg/yr is anthropogenic plus natural.

Reply:

The sentence in question refers to the importance of CH4 as a greenhouse gas, i.e. to climatic influence that is independent of the origin of emissions. We will add the proportion of natural emissions (40%) to the sentence.

Comment:

Page 3, line 13-15: this has been known for decades, and should not be presented as something new. See for example Christensen, 1993; Torn and Chapin, 1993; Whiting and Chanton, 1992.

Reply:

Our intention was not to imply that this is a new finding. We will rephrase this sentence

and consider the suggested literature (thanks for pointing these out).

Comment:

Page 5, line 27-28: a table with vegetation descriptions for each class would be helpful.

Reply:

We will add such a table.

Comment:

Page 6, line 9: have you considered NDWI as another wetness index?

Reply:

We have an accurate classification of water areas and we used the topography-based TWI to predict the location of potentially wet soil, which is relevant to CH4 production. We did not test NDWI.

Comment:

Page 10, line 23: is this normalization necessary? This paragraph seems like a complex way of simply saying that you have some unknown data in your average. In any case, at 1.4 km from the tower the effect would be negligible.

Reply:

For many calculations it is indeed necessary. Without this normalization we would underestimate the footprint-weighted averages, if the weights do no add up to 1. As shown in Fig. 3, the effect is significant especially in stable conditions.

Comment:

Page 12, line 23: which Eriophorum species? Not all of them are high emitters of methane, like Eriophorum vaginatum.

Reply:

We will add details to the revised version.

Comment:

Page 15, line 6-7: "the areal coverage of the LCC within the study area"? what do you mean? The LCC map?

Reply:

There is a typo; it should read 'LCCs'.

Comment:

Page 17, line 27: this feels like cheating. The tower is not representative for the larger region, so you reduce the region. Please don't.

Reply:

We object this kind of language in scientific correspondence. Furthermore, the comment misses the point of the discussion in this paragraph. We show that the spatial representativeness of EC measurements (sensor location bias) depends on the scale. In contrast to what the reviewer states, the tower actually is representative also on a larger scale ("the overestimation of EC measurements of the CH4 flux averaged over the study area is not statistically significant ($p > 0.05$)", p.17, l.26-27), and referring to our calculations, we simply point out the scale that would minimize the bias. We believe it would have been fully acceptable if we had originally chosen a radius of 1 km rather than 1.4 km, as there is no standard procedure for defining the study area, which potentially results in a significant and unknown measurement bias. In stark contrast to "cheating", we seek for objective criteria for a consistent definition. We will rephrase these sentences to make our point clearer.

Comment:

Page 18, line 2: there's no doubt that lakes are a significant source of methane in the Arctic. Please rewrite this sentence to remove the suggestion that it may not be (e.g.

'Nevertheless, Arctic lakes and ponds emit significant amounts of CH4 in general).

Reply:

We agree. The grammatical error will be corrected by removing 'may'.

Comment:

Page 31, figure 1. Please don't use a continuous color map for land cover. Use discrete colors.

Reply:

While the colour maps used in Figs. 1c-d are continuous (for continuous variables), this is not true for Fig. 1a that shows the discrete land cover classes.

---

## Author Response (AR1)

*This document consists of the following parts: (1) Reply to the comments presented by the Associate Editor; (2) Overview of the changes to each section of the manuscript; (3) Summary of the revisions in response to reviewers' comments. For further discussion of the comments, please see our original replies (https://doi.org/10.5194/bg-2018-155-AC1 and https://doi.org/10.5194/bg-2018-155-AC2); (4) Marked-up*

5 *manuscript indicating the changes. In addition to the revisions reported in (2) and (3), there are linguistic and stylistic corrections throughout the manuscript.*

**(1) REPLY TO ASSOCIATE EDITOR**

location bias. We found that methane ($CH_4$) fluxes varied strongly with wind direction (from –0.09 t[○]).59 µg m$^{-2}$ s$^{-1}$ on average), reflecting the distribution of different LCCs. Using footprint weights of grouped LCCs as ̶ ̶ ̶ ̶s for the measured $CH_4$ flux, we then developed a multiple regression model to estimate LCC-specific fl ̶ ̶ ̶ ̶wed that wet fen and graminoid tundra patches in locations with a high topography-based wetness in ̶ ̶ ̶ ̶ ̶ $CH_4$ sources (0.95 µg m$^{-2}$ s$^{-1}$), while mineral soils were significant sinks (–0.13 µg m$^{-2}$ s$^{-1}$). ̶ ̶ ̶ ̶ ̶ ̶the

> **merboldl** Reply ✕
> unclear whether this is C or $CH_4$?
> 20.9.2018 15:32 ↩1
>
> **tuovinen**
> 'CH4' added to all flux units

Germany) sonic anemometer/thermometer, an LI-7000 (LI-COR, Inc., Lincoln, NE, USA) $CO_2/H_2O$ analyser a ̶ ̶ ̶ ̶RMT-200 (Los Gatos Research, Inc., San Jose, CA, USA) $CH_4$ analyser. The measurement height was 3 m. T ̶ ̶ ̶ ̶mpling frequency was 10 Hz, and the turbulent fluxes were calculated with the in-house PyBARFlucCalc program ̶ ̶ ̶ ̶with 30 min block averaging according to standard procedures, including double coordinate rotation, lag determination, wet-to-dry mole block averaging according to standard procedures, including double coordinate rotation, lag determination, wet-to-dry mole fraction conversion and high-frequency flux loss correction where necessary (Aubi ̶ ̶ ̶ ̶ screened for instationarity, weak turbulence (friction velocity < 0.12 m s$^{-1}$) and ̶ ̶ ̶ ̶analysed in the present study cover the period of 5 July to 29 August 2014, which ̶ ̶ ̶ ̶that year, using the daily mean air temperature of 5 °C as the threshold.[○]

> **merboldl** Reply ✕
> ↩1
>
> **tuovinen**
> Corrected.
> 20.9.2018 14:52

> **merboldl** Reply ✕
> here a bit more info would be useful. I assume you did not permanently have these 5°C. What are the individual growing season stages and how could these affect your results etc.
> 10.9.2018 15:04 ↩1
>
> **tuovinen**
> We added air and soil temperature and LAI data and extended the site description, to better characterize the study period (Sect. 2.1.1). In addition, we estimated weekly CH4 fluxes to address temporal variation (Sect. 3.2).

The land cover classification was based on seven visually judged plant community types (PCTs) au ̶ ̶ ̶ ̶ed by two non-vegetated surfaces: (1) wet fen, (2) dry fen, (3) bog. (4) graminoid tundra, (5) flood meadow, (6) sh ̶ ̶ ̶ ̶ ̶tundra, (8) bare ground and (9) water (Mikola et al., 2018). The PCTs were identified within an area of ̶ ̶ ̶ ̶around the EC tower on the basis of an extensive vegetation and soil survey. They were verified using statistical ̶ ̶ ̶ ̶ ̶established study plots according to plant species composition and functional plant and soil attribute ̶ ̶ ̶ ̶

> **merboldl** Reply ✕
> how are these different from PFTs?
> 10.9.2018 15:08 ↩1
>
> **tuovinen**
> Here PCTs refer to the LCCs with vegetation. They are not the same thing as the PFTs, which were used for the classification of the plant material collected. We rephrased this sentence for clarity.

The land cover classification was based on seven visually judged plant community types (PCTs) augmented by two non-vegetated surfaces: (1) wet fen, (2) dry fen, (3) bog, (4) graminoid tundra, (5) flood ... tundra, (8) bare ground and (9) water (Mikola et al., 2018). The PCTs were identified ... tower on the basis of an extensive vegetation and soil survey. They were verified ... established study plots according to plant species composition and functional plant ...

[Figure]

**merboldl**    Reply ✕
how can one distinguish between wet and dry fen only visually?
3.8.2018 10:31
**tuovinen**
We included a more detailed description of the LCC characteristics, which indicates the difference between the fen types (Table 1).

1b). The internal and external classification accuracy of the land cover classification were 80 and 49 %, respectively. For details, see Mikola et al. (2018).

**merboldl**    Reply ✕
why one approach 80% and one only 49% . please expand here
3.8.2018 10:31
**tuovinen**
Definitions of 'internal' and 'external' were included.

Using non-linear regression, the LAI of vascular plants was estima... (NDVI) calculated from the reflectance data of the 2012 WorldView-2...

To demonstrate how the EC flux measurement at Tiksi is affected by surface heterogeneity, we calculated with Eq. (8) the footprint-weighted averages of the surface attributes LAI, terrain elevation and TWI, and with Eq. (10) the footprint-weighted LCC areas of the nine classes shown in Fig. 1a. ... ns of the variables that affect the footprint in a given $\theta$, i.e., $U$, ... bility conditions, for which typical parameter combinations wer... The

**merboldl**    Reply ✕
you mention above that LAI and terrain elevation are continuous variables - can you extend on this please
3.8.2018 10:40
**tuovinen**
This simply means that LAI, elevation and TWI are expressed as real (decimal) numbers, while land cover is expressed as a class (an integer or a name). We rephrased the sentence for clarity.

for a representative site description. Even if the coverage of the nine basic LCCs clearly differs from their footprint-weighted contributions (Table 4), the four classes aggregated according to their assumed $CH_4$ emission potential cover areas rather similar to those within the original study domain (Table 6). Within the regional upscaling area of 35.8 km$^2$, the strong emitters are less common, but the total flux is only ... freshwater bodies occupy a larger relative area (Tabl...

**merboldl**    Reply ✕
introducing nine classes in the begining but then in any case only using realistically 4 is rather misleading and this needs to be adresses in the revised manuscript
10.9.2018 15:29
**tuovinen**
We start from the general land cover classification (which is also used in other studies), which we apply in the present MS to analyse CH4 fluxes. For this, four aggregated classes was an appropriate choice, as it represents the different levels of CH4 exchange expected on different land cover types. To clarify, we reformulated this paragraph and, in a new section that addresses modelling uncertainties, explain that the uncertainties preclude modelling that could resolve all individual LCCs. We also replaced 'LCC-specific' by 'LCC group-specific' throughout the text, where appropriate.

The eddy covariance flux measurement technique is commonly considered to have an advantageous spatial averaging property, sometimes to the extent that it is assumed to "provide an accurate integration of the overall flux from the [heterogeneous] ecosystem" (Turner and Chapin III, 2006). Howev... universal premise, since this integration process involves differe... feature of EC measurements, which we in the present study demo... northeastern Russia. The $CH_4$ fluxes measured at Tiksi were highl... and soil wetness within the tundra landscape around the EC tower.

**merboldl**    Reply ✕
I suggest to read Hill et al. 2016 with Robert Clement on what an EC System sees and what in can not see.
3.8.2018 11:42
**tuovinen**
This is a very useful paper; thank you for pointing it out. We cite it in the revised MS in a few places, but not here. The highlighted sentence refers to the variation *within* the footprint, i.e. the representativeness problem, while Hill et al. address the region *outside* the footprint, i.e. the fluxes we do not measure (ecosystem averaging/upscaling problem). These are separate issues.

and soil wetness within the tundra landscape around the EC tower. During summer 2014, the mean bias of observations with respect to the upscaled flux varied strongly with wind direction, ranging from –200 to 200 %

merboldl      Reply ✕
above you state -200 to +400
3.8.2018 11:42    ↰1

tuovinen
That range from -200 to 400 % is for individual data points. Here we refer to the bias that is calculated from directional mean fluxes (corrected to -170...230 % in the revised MS). We added these values to Sect. 3.3.

By combining VHSR satellite imagery and footprint modelling, we could statistically estimate the contribution of the main land cover types to EC measurements. Methane emissions mainly originated from wet fen a...

**(2) OVERVIEW OF CHANGES**

**Abstract**

- updated to reflect the revisions to the main text
- language improved

**Introduction**

- presentation improved by removing secondary material
- scope of the manuscript highlighted

**Material and methods**

**Section 2.1.1**

- site description enhanced
- meteorological data for 2014 added, including a new figure (Fig. S1)

**Section 2.1.2**

- description of EC data processing methods extended
- description of chamber measurements included

**Section 2.1.3**

- table summarizing LCC characteristics added (Table 1)
- LAI dynamics and LCC validation clarified

**Section 2.1.4**

- LCC description extended
- LAI data included

**Section 2.2.1**

- removed

*Section 2.2.2 (now 2.2.1)*

- *updated to match the removal of Sect. 2.2.1*

*Sections 2.2.3–4 (now 2.2.2–3)*

- *minor changes only*

5 *Section 2.3.1*

- *some methodological details clarified to coordinate with Sect. 2.3.2*
- *measurement error estimation and weekly data analysis added*

*Section 2.3.2*

- *some methodological details clarified*

10 ***Results and discussion***

*Section 3.1*

- *shortened considerably*
- *number of upscaling reference scales reduced, Table 4 simplified*
- *presentation improved; a minor result updated*

15 *Section 3.2*

- *new results based on weekly data added, including a new figure (Fig. 6)*
- *new results on environmental controls added*
- *a substantial amount of new discussion and references added*
- *literature-based flux data corrected as a result of more thorough analysis*
20 - *a paragraph moved to a new section (Sect. 3.4)*

*Section 3.3*

- *discussion of upscaling reference scales simplified, Table 6 reduced*
- *flux sensor bias estimates corrected and enhanced*
- *measurement uncertainty included*

25 *Section 3.4*

- *new section on methodological issues*
- *previous paragraph on LCC accuracy and some text from the Conclusions adapted*
- *new material on footprint uncertainty, including a summary of validation studies*
- *new discussion on the feasibility and limitations of the statistical model employed*

**Conclusions**

- *shortened for enhanced focus on key results and implications*
- *presentation improved and results updated*

**References**

- *list updated according to the text changes*

**Tables**

- *one table added (new Table 1)*
- *one table moved to the Supplement (old Table 3)*
- *two tables modified substantially (Tables 4 and 6)*

**Figures**

- *Fig. 1: labels (a)–(d) added*
- *Fig. 1a: readability improved by changing the LCC colour codes*
- *Fig 4a–c: new right axes introduced*
- *Fig. 4d–f: removed*
- *Fig. 6: new material*
- *Fig. 7 (old Fig. 6): confidence interval included, appearance updated*

**Supplement**

- *footprint analysis added, based on previous Sects. 2.2.1 and 3.1*
- *Fig. S1: new material*
- *Fig. S2 (old S1): readability improved by changing the LCC colour codes*

*REFEREE #1*

5    MAJOR COMMENTS

1.) constant flux rates: I was extremely surprised when read-
ing the results section to find that temporal variability in flux rates is ignored, instead
a constant flux rate is assigned to each of the chosen LCC groups. This constitutes
an over-simplification of the tundra ecosystem in the first place, but also limits the
10   presented approach to extremely basic overall findings. Considering that fluxes over
a period of 8 weeks from the late growing season of an Arctic ecosystem are used,
besides short and mid-term climate variability the fluxes will be influenced by slowly
varying conditions such as e.g. thaw depth or soil temperatures. Accordingly, it cannot
be assumed that even mean flux levels for moving windows remain constant over such
15   a long period. Even more important in this context, I would strongly assume that the
different LCC groups will show a different trend in phenology in this part of the sea-
son, i.e. some may be subject to earlier senescence, and also some of them may react
more strongly to environmental stress such as water limitation, or first nights with freez-
ing temperatures. This implies that the differences/ratios between flux rates from LCC
20   groups will not be constant over time. So what is the value of providing just a single
mean flux rate per LCC group? Would the differences still be significant if short-term
temporal variability, and shifts in flux rate ratios, would be taken into account? Regard-
ing the applicability of the results, under the given circumstances the output of the flux
decomposition is of no value for any other purpose than investigating a potential sensor
25   location bias (and even here the impact is limited). To reach a broader audience, tem-
porally varying fluxes per biome with functional links to environmental controls would
have to be provided. Summarising this item, it is obvious that the chosen approach is
based on a strong simplification of the actual flux patterns. This should be discussed
in detail in a revised version of the manuscript. This discussion needs to be supple-
30   mented by a demonstration how variable measured net flux rates are over time, broken
up into the chosen LCC groups. If it cannot be proven that the ratios between flux rates
from different groups remain largely constant over time, the approach cannot be ap-
plied as is. In general, I strongly urge the authors to consider extending their approach,
so temporal variability in flux rates can be considered.

***Reply:*** *We extended the discussion of our results substantially, showing that the short-term CH4 flux variations*

*are mainly generated by footprint variations rather than other environmental controls (Sect. 3.2). To investigate*

*the seasonal trend, we performed a new analysis in which weekly data were used in addition to the full eight-*

*week period. These results were added to the manuscript, including a new figure (Fig. 6) and related discussion*

40   *(Sect. 3.2). In addition, the correlation of methane fluxes with soil temperature, friction velocity and leaf area*

*index was examined, reported and compared with previous studies in the revised version (Sect. 3.2). We also*

*compared the upscaling results obtained from the weekly data with the original estimate (Sect. 3.4).*

2.) uncertainty estimates: The manuscript misses to even discuss some essential sources of uncertainty that influence the given approach, and those few aspects that are treated (e.g. uncertainties in maps) are only covered qualitatively. Even very easy components, such as e.g. assigning an uncertainty to the input flux rates from the EC system, which is then projected onto the modelled LCC flux rates, is missing. Most importantly, there is no uncertainty estimate for the footprint approach. It is obvious that any source weight function can only be an approximation of the actual field of view of the sensor, as many footprint validation studies have shown in the past. In this study, however, footprint simulations are treated as a given fact. There are uncertainties in all the input parameters used to feed the footprint model, there are uncertainties associated with parameterizations/assumptions inherent to the footprint model, and there are uncertainties related to the methodology (e.g. horizontal homogeneity, stationary flow, and so on). For a modified version of this study, the authors need to provide a convincing concept to constrain the uncertainties in computed source weight functions, and how these influence the results obtained by the flux decomposition approach. In addition, the uncertainty concept should, as mentioned above, also involve the flux data uncertainty, and also the uncertainties inherent to the maps used in this study should be involved, and quantified.

*Reply: The error estimation procedure and the reported uncertainty estimates were explained in our original reply (bg-2018-155-AC1). We added related discussion to the revised manuscript as part of a new section dealing with methodological issues (Sect. 3.4). We also added here a paragraph on the uncertainties associated with footprint modelling, citing several validation studies that demonstrate the performance of the model used in our study. Concerning the measurement errors, we added an uncertainty estimate for the mean methane flux (Sects. 2.3.1, 3.3; Fig. 7).*

3.) Validation of flux rates: In Section 3.2, the authors include a good paragraph (p.16 ll.20ff) that supports the negative flux rates found for bare soil. As part of the line of argumentation, chamber measurements from a previous study are cited. I think it's fair to assume that this study did not only measure those 32 data points cited here for bare soil, but also other components within the Tiksi landscape. Why are those not used? Having flux chamber results for the different LCC groups would be the best way to validate that the flux composition actually produced realistic results. Also, the Tiksi flux tower has been running for several years now - why restrict this study to just 8 weeks? Why not use more data, so the database is more representative, and can also resolve temporal variability? Why not split the dataset into training and validation sets, so any finding can actually be evaluated?

*Reply: The coverage of the flux chamber measurements that are available from Tiksi was explained in our original reply (bg-2018-155-AC1). In the revised version, we include all the chamber data that we consider useful for the present study (Sect. 3.2). These data are now described in Sect. 2.1.2. The study period was*

*motivated and the validation procedure explained in our original reply (bg-2018-155-AC1). We revised the*

*Introduction to highlight the scope of the manuscript.*

4.) scope of this study: With the limitations of the chosen concept (constant fluxes)
5   as mentioned above, the authors should clearly restrict the scope of the study to an
estimate to constrain sensor location bias. I do not see any other application of their
method besides this (I would be glad to get convinced otherwise, e.g. by a thorough
discussion ..). I do not think they can claim to provide land cover specific CH4 flux rates,
since they present one set of mean flux rates for a single period of time, nothing more.
10  They also do not interpret EC data, since obviously there's no temporal variability, no
links to environmental controls, no interpretation why certain LCC groups show different
fluxes than others. What is being provided here is an extremely simplified approach to
estimate flux rates per LCC group, and check if, given these flux rates, the net fluxes
represent the emissions from a larger area (aka sensor location bias). Since there
15  is also no discussion which aspects influence the performance of this approach (e.g.
length scale of variability in terrain features, differences in flux rates between LCC
groups, footprint variability, etc), there is no way of telling if this approach could be
applied at other sites as well.

20  *Reply: We explained the rationale of the manuscript in the original reply (bg-2018-155-AC1) and clarified it in*

*the Introduction of the revised manuscript. For the revision, we performed a more detailed analysis of the LCC*

*group-specific fluxes and investigated their temporal variation and environmental responses (Sect. 3.2). We*

*added a new section that includes a discussion on the feasibility of the statistical method presented and the*

*uncertainties involved (Sect. 3.4).*

5.) a thorough discussion is simply missing! What is the implication of the findings?
How could the presented approach be used? Where are the weaknesses, which fac-
tors limit the interpretation of the findings?

30  *Reply: We added a significant amount of new discussion to the revised version (Sect. 3.2). We also added a new*

*section that deals with methodological issues, including the applicability and weaknesses of the approach*

*presented (Sect. 3.4). We reorganized the Conclusions to focus on the key implications of the study.*

GENERAL COMMENTS
35  1.) The introduction is well structured overall, and the 3 differ-
ent sets of objectives are clearly formulated. The paragraph on methane (starting p.2
l.25) is rather confusing, though, and should be revised.

*Reply: This paragraph was shortened and rephrased.*

2.) Section 2.2 needs a complete overhaul. Many sections, e.g. most parts of 2.2.1, are textbook knowledge, and do not need to be shown in detail herein. Section 2.2.2 is much too detailed for what actually needs to be described. You project a source weight function on gridded maps, and accumulate the weights of individual cells, sorted by categories, nothing more. Overall, this whole section is much too long. I suggest to revise it to the following structure: - 2.2.3 should be moved to the front - 2.2.2 should be shortened, and simplified, coming next - 2.2.1 should be discarded entirely - 2.2.4 should be moved as part of the results section

*Reply: In the revised version, Sect. 2.2.1 is removed and a shortened version is included in the Supplement. The rationale of Sect. 2.2.2 was explained in our original reply (bg-2018-155-AC1).*

3.) In Section 2.3, the ordering of the information should be revised. Many pieces of information given in 2.3.2 were needed to interpret the text in 2.3.1, for example

*Reply: We made some minor additions to Sect. 2.3.1 to improve the interpretation.*

4.) Results Section 3.1: The first part on general footprint characteristics should be removed (P1, P2). After all, what you basically state here is the obvious fact that footprint areas grow with stable stratification. The authors may move Table 3 to the appendix, and refer to it in the main text in case any reader wants to see the details, but this is clearly not part of the main story. The center part, highlighting the heterogeneity of surface characteristics within the footprint, reads well (P3-P5). The last part (P6+) should be revised - it is informative to describe a sensor location bias using the different surface characteristics, but the current format is confusing, using too many versions of a reference area (also Table 4 should be reduced).

*Reply: Most of the first part of Sect. 3.1 was removed. Table 3 was moved to the Supplement and briefly cited in the revised Sect. 3.1. In the latter part of this section, the number of different reference areas was reduced and Table 4 was simplified considerably.*

4.) p.16 l.10ff: I don't think it makes much sense to compare the Tiksi flux rates against values from other sites without also comparing environmental conditions, and the measurement approaches

*Reply: We improved the presentation of this comparison. For the rationale of the comparison, please see our original reply (bg-2018-155-AC1).*

5.) results section 3.3: It is confusing, and actually not understandable, why so many different reference areas have been used to compare the footprint LCC composition to.

This actually leaves the impression that the authors were searching for a nice configuration that can demonstrate that the EC measurements are actually well representative (e.g. p.17 l.27 ' the sensor location bias could be minimized by reducing the radius to 800– 1000 m'). What is the value in such an exercise? People who are interested in using EC data want to know how well they represent a LARGE area.

*Reply: In the revised version, we analyse fewer reference areas (Sect. 3.3, Tables 4 and 6). We also clarified the related presentation (Sects. 3.3, 4). Please see also our original reply (bg-2018-155-AC1).*

TECHNICAL COMMENTS

p.3 l. 16: I don't see a connection between spatial heterogeneity and the need for long-term measurements . . .??

*Reply: This sentence was rephrased.*

Section 2.1.1: A bit too brief. Soil types could be mentioned, and it should be mentioned that vegetation is given in a different subsection.

*Reply: This section was extended, including the suggested material.*

Section 2.1.2: The outline of the QC is too short. What exactly was done regarding instationarity, for example? How were unphysical outliers defined? And how were the gaps treated in the end?

*Reply: This section was revised, including the suggested material.*

Section 2.1.3: I suggest moving the definition of PCTs into a table. It should be mentioned that the dominant vegetation, and other characteristics, are described later in 2.1.4

*Reply: A new table summarizing the LCC properties was included (Table 1). A reference to Sect. 2.1.4 was added to Sect. 2.1.1.*

p.6 l.16: The authors should decide if they want to use LCC or PCT as a term for this classification. Using both is very confusing!

*Reply: The terminology was unified.*

Fig.4: the lower 3 panels are not necessary , since they show the same patterns as above, only normalized against the black dashed line

*Reply: The lower panels were removed, and corresponding right axes were added to the upper figures.*

*REFEREE #2*

5 The manuscript by Tuovinen et al. aims to use a detailed footprint analysis of a flux tower in Northeast Siberia to identify sensor location bias, while making use of high resolution satellite imagery. I think this study is interesting but, like the other referee, I think this paper could be a lot more effective. I agree with much of what is said in the other review, but I have a few additional comments.

First of all: if sensor location bias is your main goal, this can be done a lot easier through an analysis of the land cover classification from the satellite image. The heterogeneity of the landscape is already captured in there, and by taking the average of a hypothetical footprint area at random points within the satellite image, it can be clarified

15 which locations resemble the larger area the most. A rough estimate of the footprint would be required but this would be a lot simpler than the approach in the paper.

*Reply: Please see our original reply (bg-2018-155-AC2).*

20 As for the footprint area itself: the choice of the Korman and Meixner model is curious. There have been many advances in footprint analysis since, see for example Kljun et al (2015) who described a two-dimensional footprint model that already gives the footprint contribution for each part of the footprint. The model is freely available at http://footprint.kljun.net for Matlab, R and python.

*Reply: We added discussion on the uncertainties related to footprint modelling, citing relevant validation studies.*

*Please see also our original reply (bg-2018-155-AC2).*

If the authors had used this model, they could have simplified a large part of the meth-

30 ods, which would really help with the readability of the paper. At present, the long list of equations makes it hard to follow what the direction of the paper is, especially since many of them are not referred to later on. Like the author referee said, we don't need textbook knowledge. Sections 2.1 and 2.2 should be drastically shortened to only those equations directly relevant to the paper.

*Reply: We removed Sect. 2.2.1 and added a shortened version to the Supplement. Please see also the original*

*replies (bg-2018-155-AC1, bg-2018-155-AC2).*

Also, I don't understand why there are not more flux chamber measurements in the

40 area, but only from bare soil. The spatial heterogeneity of methane fluxes can be much better identified from direct measurements rather than the inverse method presented here.

*Reply: The coverage of the flux chamber measurements that are available from Tiksi was explained in our original reply (bg-2018-155-AC2). In the revised version, we include all the chamber data that we consider useful for the present study (Sect. 3.2). These data are now described in Sect. 2.1.2.*

5 Please, when reporting EC methane flux measurements, use nmol/m2/s rather than micrograms.

*Reply: Please see our original reply (bg-2018-155-AC2).*

10 Finally, the authors should put their research in a better context compared to existing literature. The discussion is very limited, and previous studies that have had detailed footprint analyses or emphasized the spatial variation of methane fluxes are either ignored or referenced too briefly. See for example Matthes et al (2014) and Maruschak et al (2016) for two prime examples, but also the paper by Parmentier et al. (2011) which 15 you do cite but without mentioning that it also dealt with methane flux heterogeneity. There are many more, and this needs to be recognized.

*Reply: We enhanced the discussion of our results in Sect. 3.2 considerably and added a new section (Sect. 3.4).*

*The literature considered in the manuscript was extended significantly, including the references suggested here;*

20 *especially the paper by Parmentier et al. (2011) is discussed in the revised version.*

Other comments:

Page 2, line 27: it would make more sense to mention only natural emissions, since 25 this paper focuses on that. This number of 560 Tg/yr is anthropogenic plus natural.

*Reply: The proportion of natural emissions was added to this sentence.*

Page 3, line 13-15: this has been known for decades, and should not be presented as 30 something new. See for example Christensen, 1993; Torn and Chapin, 1993; Whiting and Chanton, 1992.

*Reply: This sentence was removed for clarity.*

35 Page 5, line 27-28: a table with vegetation descriptions for each class would be helpful.

*Reply: Such a table was added (Table 1).*

Page 6, line 9: have you considered NDWI as another wetness index?

*Reply: Please see our original reply (bg-2018-155-AC2).*

Page 10, line 23: is this normalization necessary? This paragraph seems like a complex way of simply saying that you have some unknown data in your average. In any case, at 1.4 km from the tower the effect would be negligible.

*Reply: Please see our original reply (bg-2018-155-AC2).*

Page 12, line 23: which Eriophorum species? Not all of them are high emitters of methane, like Eriophorum vaginatum.

*Reply: 'e.g., E. vaginatum' was added to the text.*

Page 15, line 6-7: "the areal coverage of the LCC within the study area"? what do you mean? The LCC map?

*Reply: This was corrected to 'LCCs'.*

Page 17, line 27: this feels like cheating. The tower is not representative for the larger region, so you reduce the region. Please don't.

*Reply: We rephrased these sentences to clarify our presentation.*

Page 18, line 2: there's no doubt that lakes are a significant source of methane in the Arctic. Please rewrite this sentence to remove the suggestion that it may not be (e.g. 'Nevertheless, Arctic lakes and ponds emit significant amounts of CH4 in general).

25  *Reply: We corrected this sentence by removing 'may'.*

Page 31, figure 1. Please don't use a continuous color map for land cover. Use discrete colors.

30  *Reply: We changed the colour codes in Fig. 1a (and a corresponding LCC map in Fig. S2) to improve the readibility.*

35  *(4) MARKED-UP MANUSCRIPT*

[revised manuscript text omitted]

**(a) Land cover and mean footprint**

[Figure]

**Land cover and mean footprint**

[Figure]

[Figure]

[Figure]

[Figure]

[Figure]

[Figure]

Figure 1. Land cover classes and the mean cumulative footprint during the growing season of 2014 (a), maximum leaf area index (on 12 August 2012) (b), terrain elevation (c) and topographic wetness index for terrestrial surfaces (d). The isophlets in (a) indicate the areas with a 25, 50, 75 and 90 % contribution to the measured flux (only the further distance visible). The plus sign indicates the location of the EC tower.

[Figure]

5    Figure 2. Distribution of the aggregated land cover classes (excluding marine areas).

[Figure]

5    Figure 3. Proportion of different land cover classes in the flux footprint as a function of wind direction for the three flow condition cases specified in Table 2. The rightmost panel shows the relative coverage of these classes within the study area (%).

[Figure]

[Figure]

Figure 4. The footprint-weighted and areally averaged leaf area index (a), terrain elevation (b) and topographic wetness index (c) as a function of wind direction for the three flow condition cases specified in Table 2. The right-hand ordinate indicates the corresponding sensor location bias.

[Figure]

Figure 5. Measured and modelled $CH_4$ fluxes as a function of wind direction (left axis). The averaged data were calculated in 50 direction classes. The right axis indicates the sensor location bias of the measured data shown (both individual points and the mean) with respect to the mean upscaled flux within the study area (0.183 µg $CH_4$ m$^{-2}$ s$^{-1}$).

[Figure]

Figure 6. Estimates of the LCC group-specific fluxes calculated from weekly data (1 = 5–11 July; 2 = 12–18 July; 3 = 19–25 July, data missing; 4 = 26 July –1 August; 5 = 2–8 August; 6 = 9–15 August; 7 = 16–22 August; 8 = 23–29 August). The vertical bars indicate the 95 % confidence intervals.

[Figure]

Figure 7. Upscaled CH4 flux within a circular area as a function of the distance from the EC tower (left), and the corresponding sensor location bias according to Eq. (5) (right). The red line indicates the mean measured flux. The shaded areas represent the 95 % confidence intervals .

---

## Author Response (AR2)

*RESPONSE TO REVIEW COMMENTS*

*Referee 1*

5   I will restrict this review to the revised aspects of the newly submitted manuscript, and focus on the changes that address my major points of concern from the previous review. Overall, I acknowledge that my major areas of concern have been addressed well by the authors, so that the current version of the text has been significantly improved. I still think there is room for improvement, but this may be covered in a follow-up study that was announced by the authors in the conclusions.

10  Structure and length of the manuscript text are now well chosen, here particularly the methods section has benefitted from the revisions compared to the original manuscript. The results and their interpretation, separated into 3 major blocks, are convincingly organised, and well connected. The presentation of results in figures and tables is adequate. As the bottom line, I have only a few minor comments that should be addressed before publication of this study (see below).

15  Besides these minor things, one of my major concerns, the assessment of uncertainties and their impact on the study results, still has not been fully addressed. I acknowledge that a new section (3.4) has been added discussing methodological issues, but this is far from being a comprehensive uncertainty assessment. Most of the things that are addressed here in a qualitative fashion could as well be quantified, e.g. by running repeated optimisations based on randomly disturbed LCC maps or

20  the input parameters for the footprint model. Also, the uncertainty of the eddy fluxes and their impact on the performance of the flux partitioning should be treated. I believe that the presented manuscript could be substantially strengthened through such a quantitative uncertainty assessment; however, I am not demanding this as mandatory for the given paper. If the authors decide to leave their uncertainty treatment at a qualitative level for now, I highly recommend to include the quantitative assessment for

25  the follow-up paper.

*Reply: We note that all flux results presented in the manuscript do include quantitative uncertainty estimates. Concerning the methodological issues discussed in Section 3.4, however, we certainly agree that a more quantitative assessment, along the lines of what is suggested here, would shed more light on these issues. We*

30  *thank the reviewer for these suggestions that are beneficial for our follow-up studies.*

minor comments:

- the 4th paragraph of the introduction (p.3 ll.10ff, general intro on footprints) is not contributing much to the story. It may be shortened, and combined with the following paragraph

*Reply: We shortened this paragraph and combined it with the following paragraph.*

- move 2.3.2 further up, since 2.3.1 partly refers to it. Options would be to either either swap 2.3.1 and 2.3.2, or move this up to the landscape classification section as 2.1.5

*Reply: We swapped Sections 2.3.1 and 2.3.2 and edited the text accordingly.*

- in Section 3.2, the only part missing is that the actual LCC fluxes are not given in the text (only in a table), so that a quantitative comparison against chamber data does not make sense

*Reply: In general, it is unnecessary to repeat numbers that are presented in a table. However, we acknowledge that any reference to the LCC fluxes is missing from the paragraph reporting this comparison (Section 3.2, 4th paragraph). We added here two references to Table 5 to identify the data that are compared against the chamber data.*

- p.20, l.9ff: it is unclear how the discussion on definition of target area adds to this story (same for the conclusions)

*Reply: We believe that this is pertinent discussion that directly emerges from our analysis. Our results show that there can be a significant mismatch between the site description data (e.g., LCC proportions) and the field of view of EC measurements, i.e. the LCC composition actually observed. This is presented as a practical implication of the study.*

Given these minor edits will be implemented, I recommend the manuscript for publication.

This is my second review of the paper by Tuovinen et al. and although I feel that the manuscript has improved, I think it still needs a lot of work because there are many aspects that are simply not clear enough. What the paper aims to do is not overly complicated, but the text is still very difficult to follow from the methods section onwards. Moreover, the approach relies strongly on idealized mathematical theory with a long list of assumptions, which disregard the chaotic nature of natural ecosystems. These assumptions may be valid for the purpose of this paper but need to be better supported.

*Reply: We revised the presentation for improved clarity, as detailed below.*

Section 2.2.1 reads as a background chapter in a textbook without a clearly presented goal. The frequent use of "we can" without a clearly stated purpose does not make it obvious to the reader how these equations will be used in the rest of the paper. A reader may wonder whether this is background information or whether it's used in the final calculations. This is not clear until the reader has reached the results section. It would help to rewrite the methods section and clearly indicate at each stage what the goal is of the chosen approach, without ambiguous wording. This helps the reader to follow the paper a lot better and avoids that people will have to read it twice to understand the approach.

*Reply: We edited Section 2.2.1 (a subsection of Material and methods) to remove the obscurity noted by the reviewer. We added an introductory sentence to the beginning of this Section to highlight the fact that the equations presented constitute an integral part of the methods employed in the study. In the end of this Section, we now explicitly indicate how each equation is connected to the rest of the paper.*

The same unclarity goes for the many assumptions this paper makes in the methods section, which are not properly argued or followed up on. Here is a short list:

- In section 2.2.3 sigma_v/u* is fixed at the median value of 2.3, but what is the consequence of doing this for all stabilities? By how much does sigma_v/u* vary and does it change your results?

*Reply: sigma_v/u\* only affects turbulent diffusion in the crosswind direction. Thus its value is unimportant for those calculations in which all wind directions are weighted equally or the crosswind-integrated footprint is considered. Concerning the results presented in the manuscript, the choice of sigma_v/u\* only has an effect on the footprint dimensions of the 'Single three-dimensional' case in Table S1 in the Supplement. In addition, the stability dependence of sigma_v/u\* proved to be rather unsystematic in our data. In the footprint calculations for the statistical model, the observed sigma_v and u\* were used directly, i.e., no scaling was assumed. We added a brief note about the crosswind diffusion (Section 2.2.3).*

- Page 10, line 16: it is assumed that methane fluxes do not vary within an LCC which is very unlikely considering previously published studies. The effect of this choice is not shown.

*Reply: In principle, the fluxes of course vary within any LCC. However, the text here only implies that the model is formulated in terms of the mean fluxes and that we are interested in the differences of these means among different LCCs. Previously published studies indicate that these mean fluxes differ widely, as discussed in the manuscript. If the spatial variation within an LCC were too large, the modelling would fail. The rationale behind this approach is explained in Introduction (last paragraph, 2nd objective) and discussed in Section 3.4 (2nd paragraph).*

- Page 10, line 20: it is further assumed that LCC-specific fluxes remain constant in time even though Figure 6 shows this is not the case.

*Reply: Similarly to the previous comment, this statement relates to the model structure. The LCC-specific fluxes are not assumed to be universally constant. Instead, constancy is assumed for any arbitrary data set (e.g., representing a time period or a temperature range) that is analysed with the statistical model. For the results shown in Figure 6, the model was applied on a weekly basis, i.e., we calculated separately the mean weekly fluxes, which indeed were different from each other. We removed the reference to the number of observations in the full data set here as this may appear confusing.*

- Page 11, line 13: it is assumed that these confidence intervals reflect the overall uncertainty, but errors in ec-fluxes are highly non-linear. Is OLS still applicable in that case? Also, no evidence is presented that this is a proper replacement for a bottom-up error analysis.

*Reply: Such non-linear errors produce a heteroskedastic model error structure, which is what was observed in the present study, as reported in the manuscript. In this case, OLS is still applicable for the regression coefficient estimates, which are unbiased, but not for the variance and standard error estimates as these are biased. Therefore, we calculated the confidence intervals using a heteroskedasticity (and autocorrelation) consistent variance estimator. We have also tested a more advanced weighted least squares technique; this produced very similar results for the mean fluxes and thus was not adopted for the manuscript. A bottom-up error analysis would shed more light on the role of different error sources, but given that we have a well-defined statistical problem that is analysed with appropriate methods, the confidence intervals obtained should be considered fully justifiable in the context they are presented, i.e., for testing the hypotheses introduced and for representing the uncertainty of the mean fluxes estimated.*

- Page 11, line 21: it is assumed that model residuals represent measurement error. This doesn't make sense. These residuals indicate how well your model performs, not the other way round. Yes, there is uncertainty in ec-measurements but there are well-established methods in the community on how to calculate random and systematic measurement errors.

*Reply: This error estimate is based on the rationale that here we want to estimate the standard error of the measured mean flux (the red line in Figure 7). If the flux were constant, we would calculate this error as standard deviation/sqrt(n). As the flux varies strongly due to environmental drivers, most notably with footprint, this approach would seriously overestimate the variance related to the measurement itself. The modelled flux, which on average is unbiased, was thus introduced to remove most of the environment-driven variance and to produce a conservative estimate. As the use of modelled fluxes as reference may seem controversial, we replaced this approach by now calculating the dispersion with respect to the mean observed flux of 50 binned wind direction classes. To a close approximation, this results in the same error estimate as originally presented. We revised the description of the related methods in Section 2.3.1 (2.3.2 after reorganisation), while the results remain unchanged.*

- Page 12, line 11: it is hypothesized that the neutral group is not statistically different from zero. Is this tested? Figure 6 appears to suggest otherwise, especially in week 7.

*Reply: The model results include the 95% confidence intervals, which indicate whether the flux is statistically different from zero or not. Table 5 shows that the mean flux of the neutral group is not different from zero for the full study period. Figure 6 shows that typically this is true also for the weekly results. Week 7 is the exception, where the confidence intervals of all the LCC groups are wide, due to a poor coverage of different wind directions. This is discussed in Section 3.4.*

All these assumptions explain why there is such a huge difference in the spread between the modelled and measured values. This does not represent, as suggested by the authors, measurement error, but rather an inability of their model to represent this variation. This should be easily visible from either a scatter plot or a time-series, but this is not presented.

*Reply: The model represents 80% of the total variance in the measurements, which can be considered a good performance metric, so it is not true that there is a huge difference in the spread between the modelled and measured values. If the comment refers to the dispersion after the effect of wind direction is removed (Figure 5), then obviously the model does not represent this residual variation well, as footprint is the dominant factor determining the LCC group proportions, i.e., the explanatory variables of the model. A scatter or time series plot does not indicate anything else. Also, we do not claim that the unexplained variance is due to measurement error alone. Rather, we mention in passing that "part of this variation arose from measurement noise", Section 3.2, 2nd paragraph). For further illustration, however, we added a scatter and a time series plot to the Supplement (Figure S5), which show how well the model is able to reproduce the short-term variation.*

The lack of a correlation with soil temperature, as mentioned on page 15, line 24, and page 7, line1, may have to do with the fact that soil temperature was measured at a single point, although this is not clearly stated in the paper. If so, it's unsurprising that there is a poor correlation with spatially averaged data. Also, it's not clear to me whether this was a linear correlation? Methane fluxes vary non-linearly with temperature. Furthermore, Figure S1 clearly shows that the model and soil temperature have highly different time intervals. Does the correlation become better when soil temperature is binned according to the same intervals as the model?

*Reply: Using several measurement points would not necessarily increase correlation, as the common air temperature drives soil temperatures and variations can thus be expected to be similar at different locations.*

*From the available temperature measurements, those from a dry fen plot were considered the most representative ones. We added a clarification to the text (Section 2.1, last paragraph) and to the caption of Figure S1. The reported correlation was linear. We agree that an exponential relationship is more plausible, but such a fit produced an equally low R2. We added a note on this to the text (Section 3.2, 3rd paragraph). The temperature*
5   *correlation for weekly averages (i.e., the modelling interval in Figure 6) is discussed in the last paragraph of Section 3.2.*

But in principal, the investigation of the temporal variation in fluxes should be done at a higher frequency. Methane fluxes can vary strongly within a day and within a week, especially after rain
10  storms. However, the influence of changes in the water table on temporal trends is not discussed in this paper. The calculated TWI provides only spatial, but no temporal information.

*Reply: Our results show that the footprint-controlled variation dominates the high-frequency variation. A more detailed analysis of other controls was not an objective of this manuscript but will be presented in a forthcoming*
15  *study. The aims of the present manuscript are explicitly described in Introduction (last paragraph).*

Finally, I'd like to comment on my earlier review where I said that adjusting the study area to achieve a better model fit "felt like cheating". I can understand how this may have been perceived as an accusation, which was not my intention. I could have used less candid words, but my remark was an
20  honest response to the way this was written. If post-hoc changes are made to make the results look better, this does not look favorably. If I, as a reviewer who tries to read this paper as carefully as possible, misinterpret what's written then others will do so, too. It's important that the authors avoid such misinterpretation. In that light, the addition of the extra sentence about post-hoc interpretation is welcome. If the authors also add an extra reference to Figure 7 for clarification, it may further help to
25  avoid misinterpretation by others.

*Reply: We did not adjust the study area to achieve a better model fit. Instead, we analysed the spatial scope of EC measurements. Nevertheless, we acknowledge that the original text was imprecise and had to be revised. We also agree that a reference to Figure 7 is necessary here; it was placed just before the 'post hoc' sentence*
30  *(Section 3.3, 3rd paragraph).*

Minor comments:

The authors insist of the use of microgram/m2/s as a flux unit, since gram is an SI unit. This was not the point of my remark. CO2 fluxes measured with ec are almost always presented as micromol/m2/s and within the flux community it has become common practice to report methane fluxes in nmol/m2/s. The use of grams is a remnant following from flux chamber studies but not helpful in the context of eddy covariance unless cumulative budgets are discussed.

*Reply: That may be a common practice, but it is not obvious at all why it should be maintained. As this comment indicates, use of nmol/m2/s commonly results in an unwanted situation where two different units must be used in a single paper. Unless the Associate Editor decides otherwise, we would prefer to use grammes.*

Page 12, line 5: which other fens?

*Reply: We rephrased this to: "The drier fens within the study area…".*

Page 12, line 6: which previous data?

*Reply: We rephrased this to: "The syntheses cited above…".*

Page 14, line 4-10: the studies cited here used very different scales for their analysis. How is this a comparison?

*Reply: The reviewer probably means that our study area is larger than the 1 km2 area mentioned in the text. In both of the studies cited here, the spatial representativeness of EC tower measurements was assessed with respect to different reference scales. In the first paper (Wang et al.), the results were reported for three scales, of which we cited the smallest (1 km2) and largest (300 km2) ones to indicate that the bias was moderate even at a scale much larger than that of an EC site. Concerning the second paper (Kim et al.), we cited the 1 km2 scale, which is the only one for which the sensor location bias is reported. However, it is possible to deduce from other results presented in this paper that the sensor location bias remains approximately the same for areas of up to 4 km2*

*(the maximum range considered), which is similar to our study area. We added a note on this to the text (Section 3.1, 5th paragraph).*

Page 14, line 22: is this map derived in the same way as the smaller map?

*Reply: The smaller LCC map is a subset of the larger LCC map, as indicated in Material and methods (Section 2.1.3).*

Page 15, line 9: the inability of your model to capture high-frequency dynamics in methane fluxes should not be dismissed as 'measurement noise'

*Reply: Again, it is not true that the model is unable to capture high-frequency dynamics. As mentioned above, and reported on the previous line in the manuscript, 80% of the variance of the 30-min fluxes is captured. We also do not dismiss the unexplained variance as measurement noise but simply state that such noise is present.*

Page 20, line 2: the poor fit to the high frequency data, as shown in figure 5, shows that this method cannot be used for gapfilling of the time series at 30 min intervals. It only provides averages which may make sense at seasonal timescales, but not shorter. There are much better gapfilling methods available.

*Reply: The alleged poor fit to the high-frequency data seems to be a misunderstanding, as the model is tested against those data and shown to perform well. The explanatory skill is unlikely to be worse than that of any typical gap-filling model based on temperature response, for example. We acknowledge that the model is unable to reproduce the level of highest peaks, but that is a typical situation with any model and does not mean that the model is unable to produce half-hourly variations. Both these features can be seen from the time series plot that was added to the Supplement (Figure S5).*

[revised manuscript text omitted]